# Video Diffusion Models - A Survey

## Abstract

Diffusion generative models have recently shown great potential for video generation and editing tasks. This survey provides a systematic overview over relevant aspects of video diffusion models such as applications, architecture, and temporal dynamics. Developments in the field are outlined through paper summaries. The review concludes with an examination of remaining challenges and an outlook on the future of the field.

## 1 Introduction

Diffusion generative models (Sohl-Dickstein et al., 2015) have demonstrated a remarkable ability for learning heterogenous visual concepts and creating high quality images based on text descriptions (Rombach et al., 2022; Ramesh et al., 2022). Currently, a lot of effort is being made to also explore their potential for various video generation and editing tasks. Adapting generative diffusion models to video generation poses unique challenges that still need to be fully overcome. These challenges relate to aspects such as temporal consistency, video length, and computational costs.

In this survey, we identify relevant aspects of video diffusion models such as possible applications, the choice of architecture, mechanisms for modeling of temporal dynamics, and training modes (see Fig. 1 for an

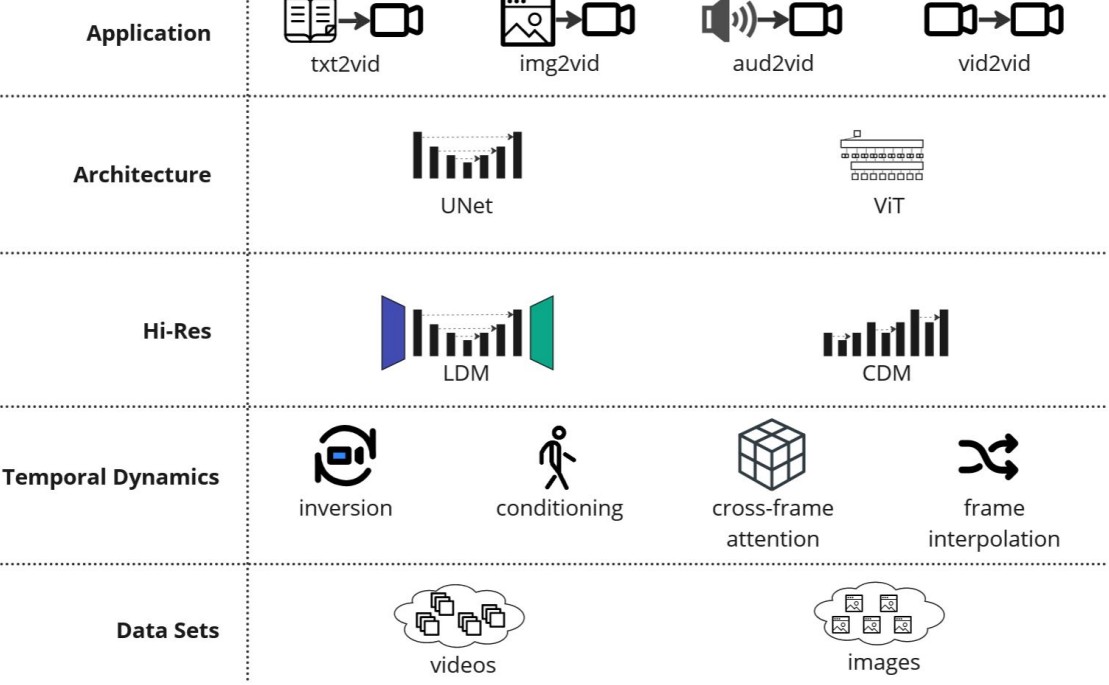

Figure 1: Overview of important aspects of video diffusion models.

overview). We then provide brief summaries of notable papers in order to outline developments in the field until now. We conclude with a discussion of ongoing challenges and try to point out potential areas for future improvements.

## 2  Taxonomy of Applications

The possible applications of video diffusion models can be roughly categorized according to input modalities. This includes text prompts, images, videos, and auditory signals. Many models also accept input that is a combination of some of these modalities. Fig. 2 visualizes the different applications. We summarize notable papers in each application domain starting from section 6.1.3. For this, we have categorized each model according to one main task.

In our taxonomy, *text-conditioned generation* (Sec. 6.1.3) refers to the task of generating videos purely based on text descriptions. Different models show varying degrees of success in how well they can model object-specific motion. We thus categorize models into two types: those capable of producing simple movements such as a slight camera pan or flowing hair, and those that can represent more intricate motion over time, such as those incorporating Physical Reasoning (Melnik et al., 2023).

In *image-conditioned video generation* (Sec. 6.4) tasks, an existing reference image is animated. Sometimes, a text prompt or other guidance information is provided. In practice, only few models offer this ability, and they are usually also specialized on a different application domain (such as *text-to-video*). For models introduced in other sections, we mention their capability for *image-to-video* generation where applicable.

We treat *video completion* (Sec. 7) models that take an existing video and extend it in the temporal domain as a distinct group, even though they intersect with the previous applications. Video diffusion models typically have a fixed number of input and output frames due to architectural and hardware limitations. To extend such models to generate videos of arbitrary length, both auto-regressive and hierarchical approaches have been explored.

*Audio-conditioned* models (Sec. 8) accept sound clips as input, sometimes in combination with other modalities such as text or images. They can then synthesize videos that are congruent with the sound source. Typical applications include the generation of talking faces, music videos, as well as more general scenes.

*Video editing* models (Sec. 9) use an existing video as a baseline from which a new video is generated. Typical tasks include style editing (changing the look of the video while maintaining the identity of objects), object / background replacement, deep fakes, and restoration of old video footage (including tasks such as denoising, colorization, or extension of the aspect ratio).

## 3  Architecture

### 3.1  Generator principle

A diffusion model for image generation implements its generation process as a chain of denoising steps that start from an input image that is a sample from a gaussian distribution of uncorrelated white noise. Each denoising step is performed by a neural network that has been trained to distort the noisy input image towards the image distribution of the target domain of the generation process. After a sufficient number of such denoising steps the image will have become transformed into a practically noise-free sample of the target domain. The key for this mechanism to succeed is a suitable training of the denoising networks. This is achieved by supervised training with input-output image pairs from reversed pairs of the inverse process, which is a chain of images starting from samples of the target domain that are iteratively transformed into samples from a gaussian distribution of uncorrelated white pixel noise by mixing at each iteration a constant proportion of uncorrelated Gaussian white noise into the pixel values.

In the following, we summarize the formalization of the unconditioned denoising process from Ho et al. (2020): The forward diffusion process follows a Markov chain that iteratively adds sampled noise to an initial input image $x_0$ over $T$ time steps. The Markov property ensures that the degraded image $x_t$ at time step $t$ only depends on the image $x_{t-1}$ in the immediately preceding step $t-1$. The distribution $q(x_t|x_{t-1})$

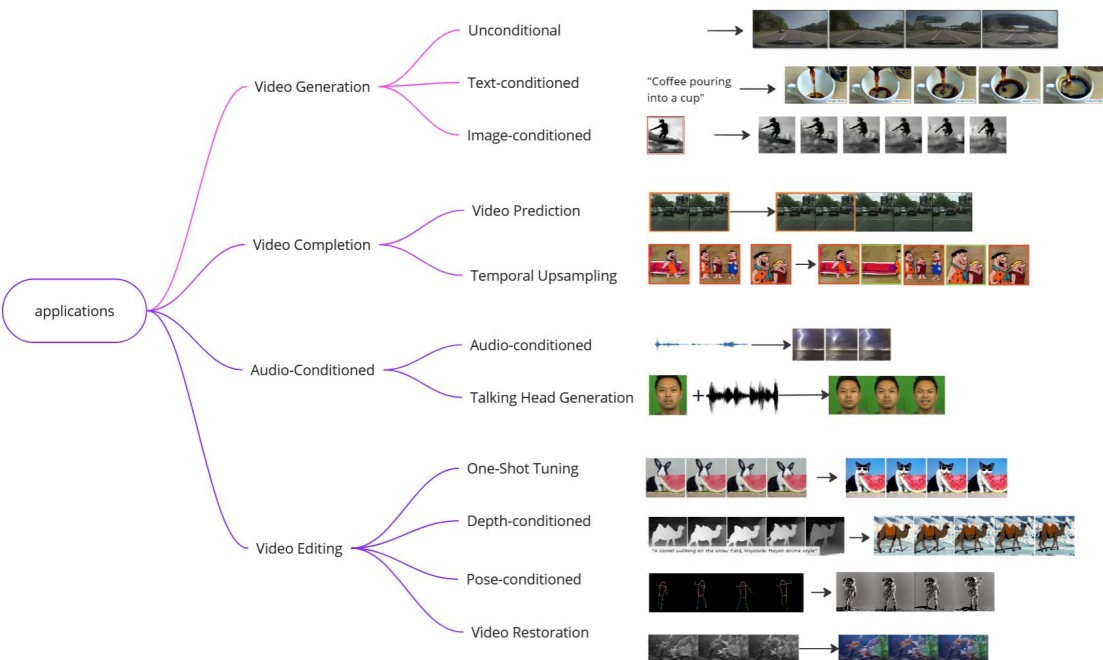

Figure 2: Applications of video diffusion models. Bounding boxes are clickable links to relevant chapters. Example images taken from following papers (top to bottom): Blattmann et al. (2023), Ho et al. (2022a), Singer et al. (2022), Lu et al. (2023), Yin et al. (2023), Lee et al. (2023b), Stypułkowski et al. (2023), Wu et al. (2022b), Xing et al. (2023), Ma et al. (2023), Liu et al. (2023a)

of $x_t$ in a forward step can be described by the Gaussian $q(x_t|x_{t-1}) := \mathcal{N}(x_t; \sqrt{1-\beta_t}x_{t-1}, \beta_t\mathbf{I})$ where the mean and standard deviation are determined by a variance-preserving noise schedule $\beta_1, ..., \beta_T$ and $\mathbf{I}$ is an identity matrix. Different schedules can be used, such as a linear or cosine schedule, influencing how quickly information in the original image is destroyed. Due to the Markov property that each state only depends on the preceding state, the overall forward process is described by $q(x_{1:T}|x_0) := \prod_{t=1}^{T} q(x_t|x_{t-1})$. Ho et al. (2020) have shown that the distribution at an arbitrary time step $t$ can be directly computed by $q(x_t|x_0) = \mathcal{N}(x_t; \sqrt{\overline{\alpha}_t}x_0, (1-\overline{\alpha}_t)\mathbf{I})$ where $\overline{\alpha}_t := \prod_{s=1}^{t} \alpha_s$ and $\alpha_t := (1-\beta_t)$.

In the denoising phase, we try to reverse this process, starting from time step $T$. The reverse process is again a Markov chain, this time with Gaussian transition probabilities that need to be learned by our model. A single denoising step is described by $p_\theta(x_{t-1}|x_t) := \mathcal{N}(x_{t-1}; \mu_\theta(x_t, t), \Sigma_\theta(x_t, t))$ where $\theta$ are the parameters of our denoising model. The full forward model is described by $p_\theta(x_{0:T}) := p(x_T) \prod_{t=1}^{T} p_\theta(x_{t-1}|x_t)$ where $p(x_T) := \mathcal{N}(x_T; 0, \mathbf{I})$.

To train the model, we minimize the variational lower bound on the negative log likelihood $\mathbb{E}[-\log p_\theta(x_0)] \leq \mathbb{E}_q \left[ -\log p(x_T) - \sum_{t>1} \log \frac{p_\theta(x_{t-1}|x_t)}{q(x_t|x_{t-1})} \right]$. This loss function can be rewritten as a sum of Kulback-Leibler divergences between the distributions of the forward and backward steps: $L := \mathbb{E}_q \left[ D_{KL}(q(x_T|x_0)\|p(x_T)) + \sum_{t>1} D_{KL}(q(x_{t-1}|x_t, x_0)\|p_\theta(x_{t-1}|x_t)) - \log(p_\theta(x_0|x_1)) \right]$. This formulation has the advantage that we can calculate closed-form solutions for the Kulback-Leibler terms. Note that the forward posteriors are now also conditioned on the initial image $x_0$. Using Bayes theorem, it can be shown that

$$q(x_{t-1}|x_t, x_0) = \mathbb{N}(x_{t-1}; \tilde{\mu}_t(x_t|x_0), \tilde{\beta}_t\mathbf{I}) \text{ where } \tilde{\mu}_t(x_t, x_0) := \frac{\sqrt{\overline{\alpha}_{t-1}}\beta_t}{1-\overline{\alpha}_t}x_0 + \frac{\alpha_t(1-\overline{\alpha}_{t-1})}{1-\overline{\alpha}_t}x_t \text{ and } \tilde{\beta}_t := \frac{1-\overline{\alpha}_{t-1}}{1-\overline{\alpha}_t}\beta_t.$$

Ho et al. (2020) showed that predicting the added noise $\epsilon_\theta(x_t, t)$ rather than the mean $\tilde{\mu}_\theta(x_t, t)$ of each forward step leads to a simplified loss function $L_{\text{simple}} := \mathbb{E}_{t,x_0,\epsilon} \left[ \|\epsilon - \epsilon_\theta(x_t, t)\|^2 \right]$ that performs better in practice.

This original formulation of the generation process as denoising diffusion probabilistic models (DDPM, Ho et al. 2020) in the form of a reverse Markov chain has more recently become complemented by a non-Markovian alternative denoted as denoising diffusion implicit models (DDIM, Song et al. 2020), which offers a deterministic and more efficient generation process. Here, a backward denoising step can be computed with $x_{t-1} = \sqrt{\overline{\alpha}_{t-1}}(\frac{x_t - \sqrt{1-\overline{\alpha}_t}\epsilon_\theta(x_t,t)}{\sqrt{\overline{\alpha}_t}} + \sqrt{1-\overline{\alpha}_{t-1}}\epsilon_\theta(x_t,t)$. One distinct advantage of this formulation of the denoising process is that it allows for accurate reconstruction of the original input image $x_0$ from the noise at time step $T$. This technique called DDIM inversion can be utilized for applications such as image and video editing (see Section 9).

### 3.2 UNet

The UNet (Ronneberger et al., 2015) is currently the most popular architectural choice for the denoising steps in diffusion models (see Fig. 3). Originally developed for medical image segmentation, it has more recently been successfully adapted for generative tasks in the image, video, and audio domains. An UNet transforms its input image into an output image of the same size and shape by encoding its input first into increasingly lower spatial resolution latent representations while increasing the number of feature channels while progressing through a fixed number of encoding layers. Then, the resulting latent representation is upsampled back to its original size through the same number of decoding layers. While the original UNet (Ronneberger et al., 2015) only used ResNet blocks, most diffusion models interleave them with Vision Transformer blocks in each layer. The ResNet blocks mainly utilize 2D-Convolutions, while the Vision Transformer blocks implement spatial self-attention, as well as cross-attention. This happens in a way that allows to condition the generative process on additional information such as text prompts. Layers of the same resolution in the encoder and decoder part of the UNet are connected through residual connections only. To train a UNet on image generation tasks, small amounts of Gaussian noise are added to the input images. The learning objective is make the UNet predict an estimate of the original input, i.e. to minimize the discrepancy between the denoised image predicted by the UNet and the input image. By letting the UNet predict and remove small amounts of noise in an iterative fashion, the network can iteratively restore pure random inputs into images from the domain that was used to create the training data. By conditioning the denoising step on encoded text prompts, the generation process can be made steerable through language input, e.g. to obtain image instances that are compatible with a specific subject-matter. It turns out that for video generation the basic UNet architecture needs to be suitably adapted, which will be the topic of section 4.

### 3.3 Vision Transformer

The Vision Transformer (ViT) (Dosovitskiy et al., 2020) is an important building block of generative diffusion models. It is a form of neural network based on the transformer architecture developed for natural language processing (Vaswani et al., 2017). Therefore, it similarly combines normalization layers, a multi-head attention layer, skip connections, as well as a linear projection layer to transform a vector of input tokens into a vector of output tokens. In the image case, the input tokens are obtained by dividing the input image into regular patches and using an image encoder to compute for each patch a patch embedding, supplemented with position embeddings to obtain a vector of input tokens. Within the attention layer, the patch embeddings are projected through trainable projection matrices, producing so called Query, Key and Value matrices. The first two matrices are used to compute a learnable affinity matrix A between different image token positions, which is calculated according to the scaled dot-product attention formula: $A(Q,K) = \text{softmax}(\frac{QK^T}{\sqrt{d_k}})$. Here, $Q$ and $K$ are $d \times d_k$ dimensional and refer to the query and key matrix, $d$ is the number of input tokens, $d_k$ the dimensionalities of the $d$ query and key vectors making up the rows of $K$ and $Q$, and the matrix $Z$ of output embeddings is obtained as $Z = AV$, i.e. the attention-weighted superposition of the rows of the value matrix $V$ (with one row for each input token embedding). In the simplest case, there is a single ($d \times d$ dimensional) affinity matrix, resulting from a single set of projection matrices. In multi-head attention, a stack of such projections with separate query, key, and value matrices is used. Their outputs are concatenated and transformed through a linear output layer to form a single set of $d$ new patch embeddings. The attention heads can be computed in parallel and allow the model to focus on

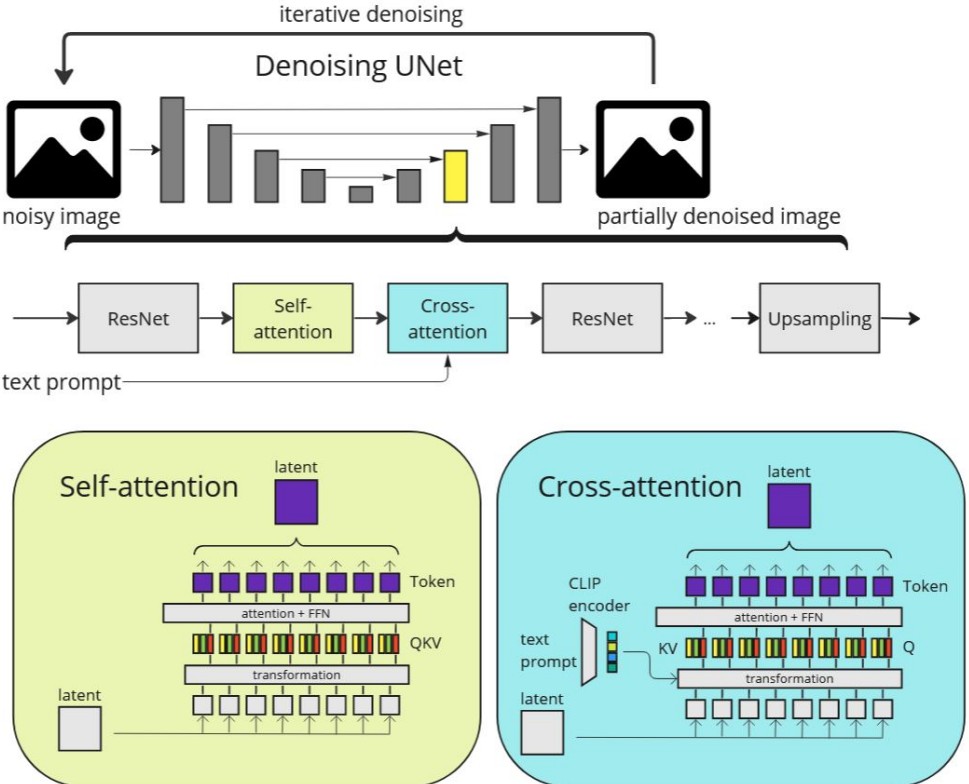

Figure 3: The denoising UNet architecture typically used in text-to-image diffusion models. The model iteratively predicts a denoised version of the noisy input image. The image is processed through a number of encoding layers and the same number of decoding layers that are linked through residual connections. Each layer consists of ResNet blocks implementing convolutions, as well as Vision Transformer self-attention and cross-attention blocks. Self-attention shares information across image patches, while cross-attention conditions the denoising process on text prompts.

multiple aspects of the image. Depending on the task, ViTs can output an image embedding or be equipped with a classification head.

In diffusion models, ViT blocks serve two purposes: On the one hand, they implement spatial self-attention where $Q$, $K$, and $V$ refer to image patches. This allows information to be shared across the whole image, or even an entire video sequence. On the other hand, they are used for cross-attention that conditions the denoising process on additional guiding information such as text prompts. Here, $Q$ is an image patch and $K$ and $V$ are based on text tokens that have been encoded into an image-like representation using a CLIP encoder (Radford et al., 2021).

Purely Vision Transformer-based diffusion models have been proposed as an alternative to the standard UNet (Peebles & Xie, 2022; Lu et al., 2023). Rather than utilizing convolutions, the whole model consists of a series of transformer blocks only. Even though far less commonly used so far, this approach might have distinct advantages, such as more flexibility in regard to the length of the generated videos. While UNet-based models typically generate output sequences of a fixed length, transformer models can auto-regressively predict tokens in sequences of relatively arbitrary length.

### 3.4 Cascaded Diffusion Models

Cascaded Diffusion Models (CDM, Ho et al. 2022b) consist of multiple UNet models that operate at increasing image resolutions. By upsampling the low-resolution output image of one model and passing it as input to

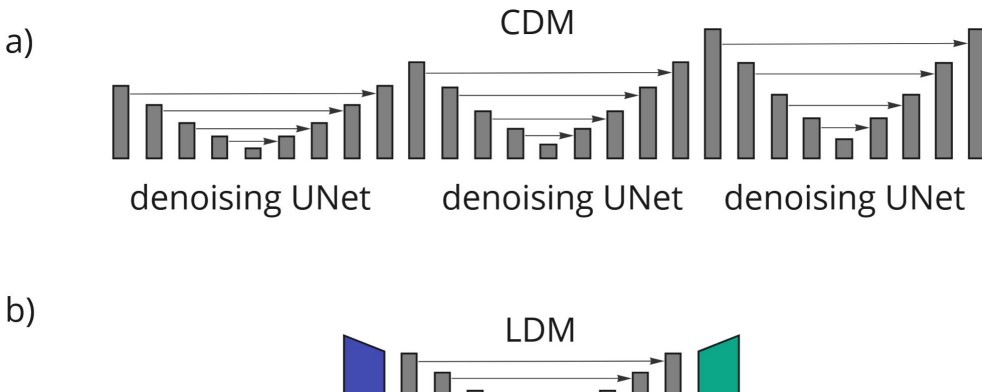

Figure 4: Architectural choices for increasing the output resolution of image diffusion models. a) Cascaded Diffusion Models (CDM) chain denoising UNets of increasing resolution to generate high-fidelity images. b) Latent Diffusion Models (LDM) use a pre-trained variational auto-encoder (VAE) to operate in lower-dimensional space, thus preserving computational resources.

the next model, a high-fidelity version of the image can be generated. At time of training, various forms of data augmentation are applied to the outputs of one denoising UNet model before it is passed to as input to the next model in the cascade. These include Gaussian blurring, as well as premature stopping of the denoising process (Ho et al., 2022b). The use of CDMs has largely vanished after the adaptation of Latent Diffusion Models (Rombach et al., 2022) that allow for native generation of high-fidelity images with limited resources.

### 3.5 LDM

Latent Diffusion Models (LDM) (Rombach et al., 2022) have been an important development of the base UNet architecture that now forms the de-facto standard for image and video generation tasks. Instead of operating in RGB space, the input image is first encoded into a latent representation with lower spatial resolution and more feature channels using a pre-trained variational auto-encoder (VAE). This low-resolution representation is then passed to the UNet where the whole diffusion and denoising process takes place in latent space of the VAE encoder. The denoised latent is then decoded back to the original pixel space using the decoder part of the VAE. By operating in a lower-dimensional latent space, LDMs can save significant computational resources, thus allowing them to generate higher-resolution images compared to previous diffusion models. Stable Diffusion [1] is an open source implementation of the LDM architecture. Further improvements of the LDM architecture have been introduced by Chen et al. (2020), who addressed specific concerns about how to adjust the architecture for high-resolution images, and Podell et al. (2023), who used a second refiner network for improving the sample quality of generated images.

## 4 Temporal Dynamics

Text-to-image models such as Stable Diffusion can produce realistic images, but extending them for video generation tasks is not trivial. If we try to naively generate individual video frames from a text prompt, the resulting sequence has no spatial or temporal coherence (see Fig. 5a). For video editing tasks, we can extract spatial cues from the original video sequence and use it to condition the diffusion process. In this way, we can produce fluid motion of objects, but temporal coherence still suffers due to changes in the finer texture of objects (see Fig. 5b). In order to achieve spatio-temporal consistency, video diffusion models need

---

[1]https://github.com/Stability-AI/stablediffusion

to share information across video frames. The most obvious way to achieve this is to add a third temporal dimension to the denoising model. ResNet blocks then implement 3D convolutions, while self-attention blocks are turned into full cross-frame attention blocks (see Fig. 6). This type of full 3D architecture is however associated with very high computational costs.

To lower the computational demands of video UNet models, different approaches have been proposed (see Fig. 7): 3D convolution and attention blocks can be factorized into spatial 2D and temporal 1D blocks. The temporal 1D modules are often inserted into a pre-trained text-to-image model. Additionally, temporal upsampling techniques are often used to increase motion consistency. In video-to-video tasks, pre-processed video features such as depth estimates are often used to guide the denoising process. Finally, the type of training data and training strategy has a profound impact on a model's ability to generate consistent motion.

### 4.1 Spatio-Temporal Attention Mechanisms

In order to achieve spatial and temporal consistency across video frames, most video diffusion models modify the self-attention layers in the UNet model. These layers consist of a vision transformer that computes the affinity between a query patch of an image and all other patches in that same image. This basic mechanism can be extended in several ways (see Wang et al. 2023b for a discussion): In temporal attention (Hong et al., 2022; Singer et al., 2022), the query patch attends to patches at the same location in other video frames. In full spatio-temporal attention (Zhang & Agrawala, 2023), it attends to all patches in all video frames. In causal attention, it only attends to patches in all previous video frames. In sparse causal attention (Wu et al., 2022b), it only attends to patches in a limited number of previous frames, typically the first and immediately preceding one. The different forms of spatio-temporal attention differ in how computationally demanding they are and how well they can capture motion. Additionally, the quality of the produced motion greatly depends on the used training strategy and data set.

### 4.2 Temporal Upsampling

Generating long video sequences in a single batch often exceeds the capacity of current hardware. While different techniques have been explored to reduce the computational burden (such as sparse causal attention,

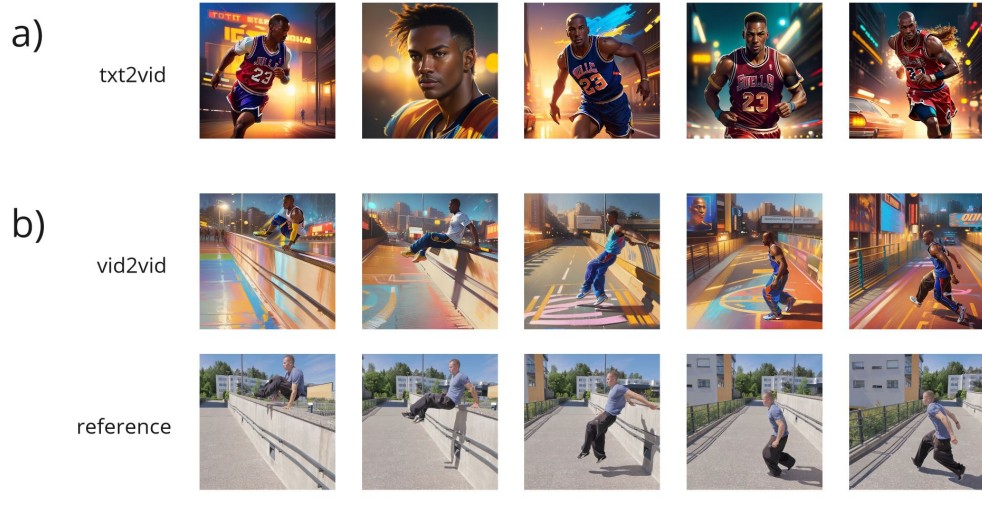

Figure 5: Limitations of text-to-video diffusion models for generating consistent videos. a) When using only a text prompt ("Michael Jordan running"), both the appearance and position of objects change wildly between video frames. b) Conditioning on spatial information from a reference video can produce consistent movement, but the appearance of objects and the background still fluctuates between video frames.

Wu et al. 2022b), most models are still limited to generating video sequences that are no longer than a few seconds even on high-end GPUs. To get around this limitation, many authors have adapted a hierarchical upsampling technique whereby they first generate spaced out key frames. The intermediate frames can then be filled in by either interpolating between neighboring key frames, or using additional passes of the diffusion model conditioned on two key frames each. It should be noted that even with this method, current diffusion models are rarely able to produce videos that are longer than a few seconds.

As an alternative to temporal upsampling, the generated video sequence can also be extended in an auto-regressive manner (Blattmann et al., 2023). Hereby, the last generated video frame(s) of the previous batch

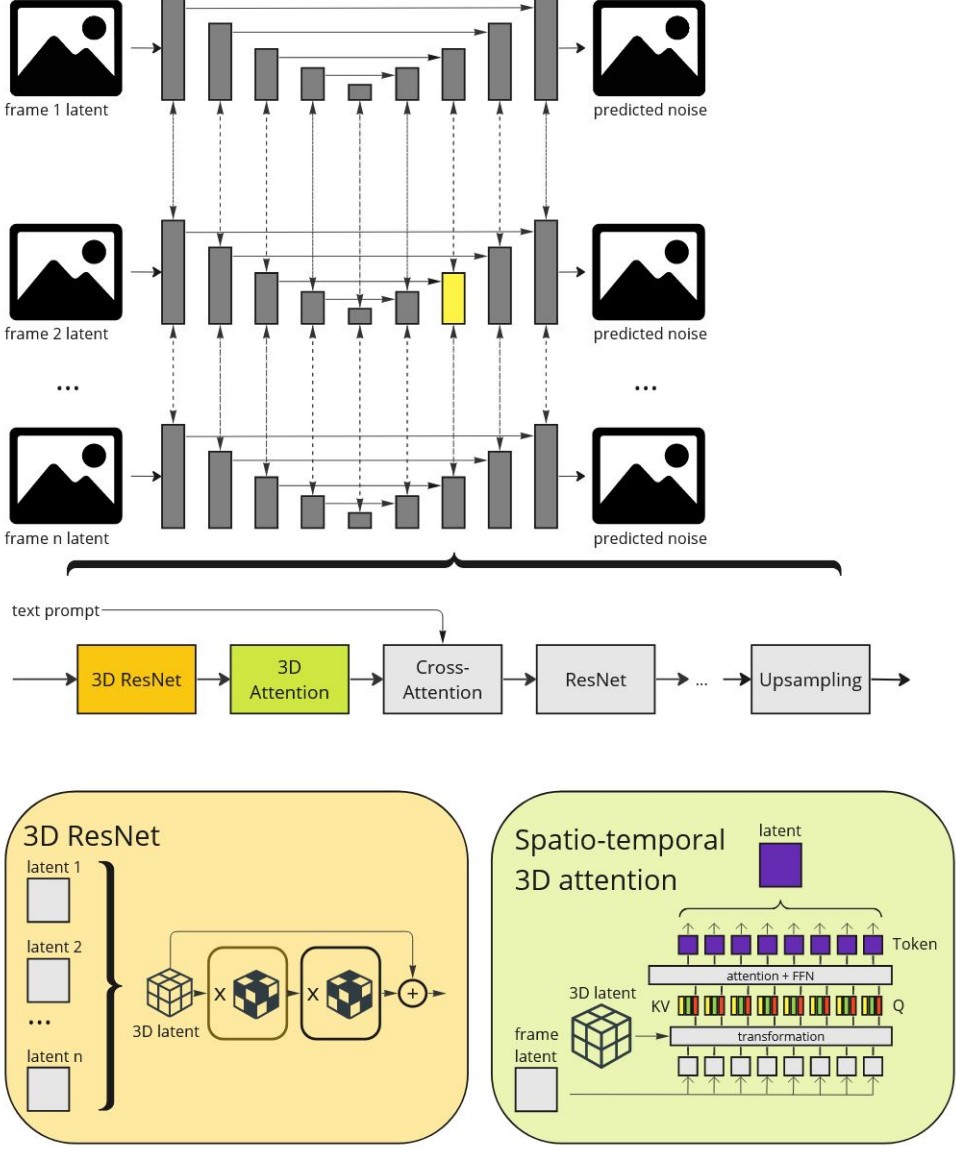

Figure 6: Three-dimensional extension of the UNet architecture for video generation. Topmost: temporally adjacent UNet 2D-layer outputs are stacked to provide 3D input at each new resolution (yellow) in the UNet layer chain. Below: processing inside the layer group starts with 3D operations, followed by cross-attention to accomodate text input, followed by flattening back to purely spatial ResNet and upsampling stages.

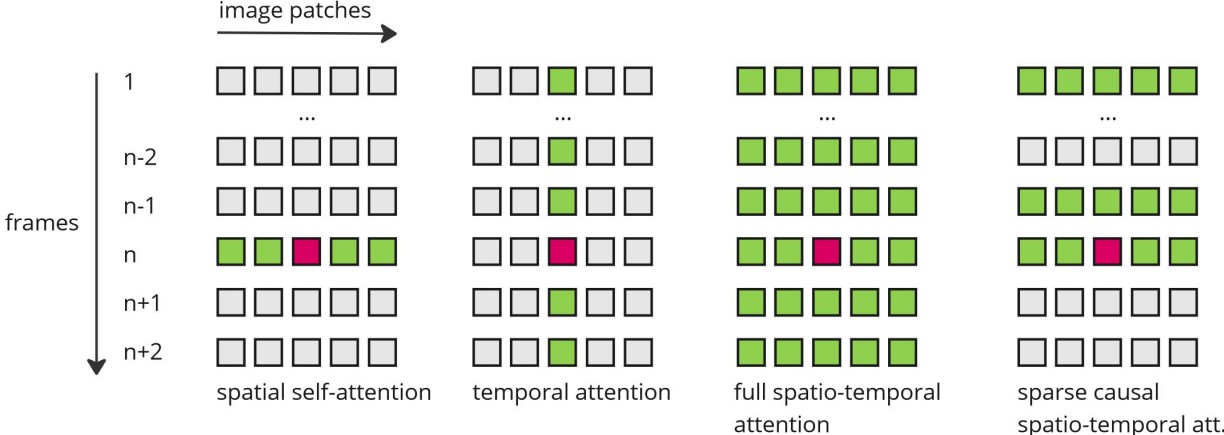

Figure 7: Attention mechanisms for modeling temporal dynamics.

are used as conditioning for the first frame(s) of the next batch. While it is in principle possible to arbitrarily extend a video in this way, the results often suffer from repetition and quality degradation over time.

### 4.3 Structure Preservation

Video-to-video translation tasks typically strive for two opposing objectives: Maintaining the coarse structure of the source video on the one hand, while introducing desired changes on the other hand. Adhering to the source video too much can hamper a model's ability to perform edits, while strolling too far away from the layout of the source video allows for more creative results but negatively impacts spatial and temporal coherence.

A common approach for preserving the coarse structure of the input video is to replace the initial noise in the denoising model with (a latent representation of) the input video frames (Wu et al., 2022b). By varying the amount of noise added to each input frame, the user can control how closely the output video should resemble the input, or how much freedom should be granted while editing it. In practice, this method in itself is not sufficient for preserving the more fine-grained structure of the input video and is therefore usually augmented with other techniques. For one, the outlines of objects are not sufficiently preserved when adding higher amounts of noise. This can lead to unwanted object warping across the video. Furthermore, finer details can shift over time if information is not shared across frames during the denoising process.

These shortcomings can be mitigated to some degree by conditioning the denoising process on additional spatial cues extracted from the original video. For instance, specialized diffusion models that have been trained to take into account depth estimates[2] can be used. ControlNet (Zhang & Agrawala, 2023) is a more general extension for Stable Diffusion that enables conditioning on various kinds of information, such as depth maps, OpenPose skeletons, or lineart. A ControlNet model is a fine-tuned copy of the encoder portion of the Stable Diffusion denoising UNet that can be interfaced with a pre-trained Stable Diffusion model. Image features are extracted using a preprocessor, encoded through a specialized encoder, passed through the ControlNet model, and concatenated with the image latents to condition the denoising process. Multiple ControlNets can be combined in an arbitrary fashion. Several video diffusion models have also implement video editing that is conditioned on extracted frame features such as depth (Ceylan et al. 2023; Esser et al. 2023; Xing et al. 2023, see Sec. 9.2.1) or pose estimates (Ma et al. 2023; Zhao et al. 2023, see Sec. 9.2.2).

---

[2]https://huggingface.co/stabilityai/stable-diffusion-2-depth

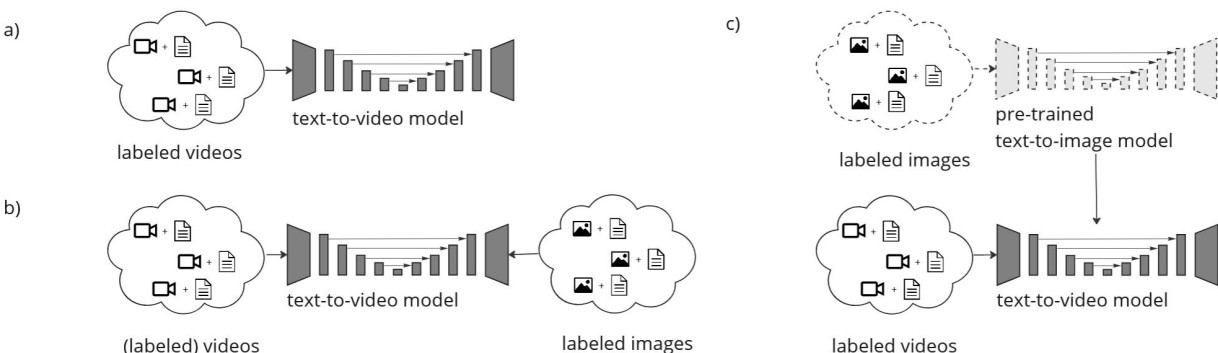

Figure 8: Training approaches for video diffusion models: a) Training on videos. b) Simultaneous training on images and videos. c) Pre-training on images and fine-tuning on videos.

# 5 Training & Evaluation

Video diffusion models can differ greatly in regards to how they are trained. Some models are trained from scratch (e.g. Ho et al. 2022c, Singer et al. 2022, Ho et al. 2022a), while others are built on top of a pre-trained image model (e.g. Zhou et al. 2022, Khachatryan et al. 2023, Blattmann et al. 2023). It is possible to train a model completely on labeled video data, whereby it learns associations between text prompts and video contents as well as temporal correspondence across video frames (e.g. Ho et al. 2022c). However, large data sets of labeled videos (e.g. Bain et al. 2021, Xue et al. 2022, see Section 5.1) tend to be smaller than pure image data sets and may include only a limited range of content. Additionally, a single text label per video may fail to describe the changing image content across all frames. At a minimum, automatically collected videos need to be divided into chunks of suitable length that can be described with a single text annotation and that are free of unwanted scene transitions, thereby posing higher barriers for uncurated or weakly curated data collection. For that reason, training is often augmented with readily available data sets of labeled images (e.g. Russakovsky et al. 2015, Schuhmann et al. 2022, see Section 5.2). This allows a given model to learn a broader number of relationships between text and visual concepts. Meanwhile, the spatial and temporal coherence across frames can be trained independently on video data that may even be unlabeled (Zhou et al., 2022).

In contrast to models that are trained from scratch (e.g. Ho et al. 2022c, Singer et al. 2022, Ho et al. 2022a), recent video diffusion approaches (e.g. Zhou et al. 2022, Khachatryan et al. 2023, Blattmann et al. 2023) often rely on a pre-trained image generation model such as Stable Diffusion (Rombach et al. 2022). These models show impressive results in the text-to-image (Rombach et al., 2022; Ramesh et al., 2022) and image editing domains (Brooks et al., 2023; Zhang & Agrawala, 2023), but are not built with video generation in mind. For this reason, they have to be adjusted in order to yield results that are spatially and temporally coherent. One possibility to achieve this is to add new attention blocks or to tweak existing ones so that they model the spatio-temporal correspondence across frames. Depending on the implementation, these attention blocks either re-use parameters from the pre-trained model, are fine-tuned on a training data set consisting of many videos, or only on a single input video in the case of video-to-video translation tasks. During fine-tuning, the rest of the pre-trained model's parameters are usually frozen in place. The different training methods are shown in Fig. 8.

## 5.1 Video Data Sets

Table 1 offers an overview of commonly used video data sets for training and evaluation of video diffusion models.

**WebVid-10M** (Bain et al., 2021) is a large data set of text-video pairs scraped from the internet that covers a wide range of content. It consists of 10.7 million video clips with a total length of about 52,000 hours. It is an expanded version of the WebVid-2M data set, which includes 2.5 million videos with an

Table 1: Video Data Sets.

| Data Set | Resolution | Source | Labels | # Clips | Clip Length | Total Length (h.) |
|---|---|---|---|---|---|---|
| WebVid-10M (2021) | - | Web | Alt-Text | 10.7 mio. | 18 sec. | 52k |
| HD-Villa-100M (2022) | 1280×720 | Youtube | Transcription (auto.) | 100 mio. | 13.4 sec. | 371k |
| Kinetics-600 (2018) | - | Youtube | Action Class | 500,000 | 10 sec. | 1,400 |
| UCF101 (2012) | 320×240 | Youtube | Action Class | 13k | 7 sec. | 27 |
| MSR-VTT (2016) | - | Web | Annotation (human) | 10k | 10-30 sec. | 41.2 |
| Sky Time-lapse (2018) | 640×360 | Youtube | - | 35k | 32 frames | 10 |
| Tai-Chi-HD (2019) | 256×256 | Youtube | - | 3k | 128-1024 frames | - |

average length of 18 seconds and a total play time of 13,000 hours. Each video is annotated with an HTML Alt-text which normally serves the purpose of making it accessible to vision-impaired users. The videos and their Alt-texts have been selected based on a filtering pipeline similar to that proposed in Sharma et al. (2018). This ensures that the videos have sufficiently high resolution, normal aspect ratio, and lack profanity. Additionally, only well-formed Alt-text that is aligned with the video content is selected (as judged by a classifier). WebVid-10M is only distributed in the form of links to the original video sources, therefore it is possible that individual videos that have been taken down by their owners are no longer accessible.

**HD-Villa-100M** (Xue et al., 2022) contains over 100 million short video clips extracted from about 3.3 million videos found on Youtube. The average length of a clip is 13.4 seconds with a total run time of about 371.5 thousand hours. All videos have a high-definition resolution of $1280 \times 720$ pixels and are paired with automatic text transcriptions. Along with WebVid-10M, HD-Villa-100M is one of the most popular training data sets for generative video models.

**Kinetics-600** (Carreira et al., 2018) contains short Youtube videos of 600 distinct human actions with their associated class labels. Each action class is represented by more than 600 video clips that last around 10 seconds. In total, the data set contains around 500,000 clips. The data set expands upon the previous Kinetics-400 (Kay et al., 2017) data set.

**UCF101** (Soomro et al., 2012) is a data set of videos showing human actions. It contains over 13,000 Youtube video clips with a total duration of about 27 hours and an average length of 7 seconds. It expands upon the previous UCF50 (Reddy & Shah, 2013) data set, which includes only roughly half as many video clips and action classes. The clips have a resolution of $320 \times 240$ pixels. Each video has been annotated with a class label that identifies it as showing one of 101 possible actions. The 101 action classes are more broadly categorized into 5 action types: Human-Object Interaction, Body-Motion Only, Human-Human Interaction, Playing Musical Instruments, and Sports. While UCF101 was mainly intended for training and evaluating action classifiers, it has also been adopted as a benchmark for generative models. For this, the class labels are often used as text prompts. The generated videos are then usually evaluated using IS, FID, and FVD metrics.

**MSR-VTT** (Xu et al., 2016) includes about 10,000 short video clips from over 7,000 videos with a total run time of about 41 hours. The videos were retrieved based on popular video search queries and filtered according to quality criteria such as resolution and length. Each clip was annotated by 20 different humans with a short text description, yielding 200,000 video-text pairs. The data set was originally intended as a benchmark for automatic video annotation but has been used for evaluating text-to-video models as well. For this, CLIP text-similarity, FID, and FVD scores are usually reported.

**Sky Time-lapse** (Xiong et al., 2018) is a collection of unlabeled short clips that contain time-lapse shots of the sky. The videos have been taken from Youtube and divided into smaller non-overlapping segments. Each clip consists of 32 frames of continuous video at a resolution of $640 \times 360$ pixels. The clips show the sky at different times of day, under different weather conditions, and with different scenery in the background. The data set can serve as a benchmark for unconditional video generation or video prediction. In particular, it allows one to assess how well a given generative video model is able to replicate complex motion patterns of clouds and stars.

**Tai-Chi-HD** (Siarohin et al., 2019) contains over 3,000 unlabeled clips from 280 tai chi Youtube videos. The videos have been split into smaller chunks that range from 128 to 1024 frames and have a resolution of

Table 2: Image Data Sets.

| Data Set | # Images | Annotation | Labels | # Classes |
|---|---|---|---|---|
| ImageNet-21k (2015) | 14 mio. | Human | Class | 20,000 |
| ImageNet-1k (2015) | 1.28 mio. | Human | Class | 1,000 |
| MS-COCO (2014) | 328k | Human | Class | 91 |
| LAION-5B (2022) | 5.58 mio. | Automated | Text | - |

256×256 pixels. Similar to Sky Time-lapse, Tai-Chi-HD can be used for training and evaluating unconditional generation or video prediction.

## 5.2 Image Data Sets

Video models are sometimes jointly trained on image and video data. Alternatively, they may extend a pre-trained image generation model with temporal components that are fine-tuned on videos. Table 2 provides a brief overview over commonly used labeled image data sets.

**ImageNet** (Russakovsky et al., 2015) is a data set developed for the ImageNet Large Scale Visual Recognition Challenge that was held annually between 2010 and 2017. Since 2012,the same data set has been used for the main image classification task. ImageNet-21k is a large collection of over 14 million images that have been annotated by humans with one object category label. Overall, there are 20,000 different object classes present in the data set that are hierarchically organized according to the WordNet (Fellbaum, 1998) structure. A subset of this dataset used for the ImageNet competition itself is often called ImageNet-1k. It contains over 1 million images that each have been annotated by humans with one object category label and a corresponding bounding box. There are only 1.000 object categories in this data set.

**MS-COCO** (Lin et al., 2014) has originally been developed as a benchmark data set for object localization models. It contains over 300.000 images containing 91 different categories of everyday objects. Every instance of an object is labeled with a segmentation mask and a corresponding class label. Overall, there are about 2.5 million instances of objects in this data set.

**LAION-5B** (Schuhmann et al., 2022) is a very large public collection of 5.58 billion text-image pairs that can be found on the internet. Access is provided in the form of a list of links. To ensure a minimal level of correspondence between the images and their associated alt-texts, the pairs have been filtered by following method: Images and texts have both been encoded through a pre-trained CLIP (Radford et al., 2021) model and pairs with a low cosine CLIP similarity have been excluded. To train image or video models, often only the subset of LAION-5B that contains English captions is used. It contains 2.32 billion text-image pairs and is referred to as LAION-2B. Additionally, labels for *not safe for work* (NSFW), watermarked, or toxic content are provided based on automated classification. The LAION-5B data set offers are relatively low level of curation, but its sheer size has proven very valuable for training large image and video models.

## 5.3 Evaluation Metrics

**Human ratings** are the most important evaluation method for video models since the ultimate goal is to produce results that appeal to our aesthetic standards. To demonstrate the quality of a new model, subjects usually rate its output in comparison to an existing baseline. Subjects are usually presented pairs of generated clips from two different video models. They are then asked to indicate which of the two examples they prefer in regard to a specific evaluation criterion. Depending on the study, the ratings can either purely reflect the subject's personal preference, or they can refer to specific aspects of the video such as temporal consistency and adherence to the prompt. Humans are very good at judging what "looks natural" and identifying small temporal inconsistencies. The downsides of human ratings include the effort and time needed to collect large enough samples, as well as the limited comparability across studies. For this reason, it is desirable to also report automated evaluation metrics.

**CLIP cosine similarity** is often used to measure prompt and frame consistency. CLIP (Radford et al., 2021) is a family of vision transformer auto-encoder models that can project image and text data into a shared embedding space. During training, the distance between embedded images and their associated text labels is minimized. Thereby, visual concepts are represented close to words that describe them. The similarity between CLIP embeddings is typically measured through their cosine distance. A value of 1 describes identical concepts, while a value of 0 implies completely unrelated concepts. In order to determine how well a video sequence adheres to the text prompt used to generate or edit it, the average similarity between each video frame and the text prompt is calculated (prompt consistency, Esser et al. 2023). In a similar fashion, it is also possible to get a rough measure of temporal coherence by computing the mean CLIP similarity between adjacent video frames in a sequence (frame consistency, Esser et al. 2023). In video editing tasks, the percentage of frames with a higher prompt consistency score in the edited over original video is also sometimes reported (frame accuracy, Qi et al. 2023).

**Inception Score** (IS, Salimans et al. 2016) is applicable to generative models trained on data sets with categorical labels. An Inception Net (Szegedy et al., 2016) classifier pre-trained on the ImageNet data set (Deng et al., 2009) is used to predict the class labels of each generated image. The IS score is then expressed by the Kullback-Leibler distance between the conditional class probability distribution $p(y|G(x))$ and the marginal class distribution $p(y)$ of the generated samples. While IS aligns well with human ratings and possesses good discriminative power (Borji, 2019), it is susceptible to noise, as shown by Heusel et al. (2017). It should be noted that IS only assesses the quality of individual images. When applied to video data, it can therefore not take into account aspects such as temporal coherence between video frames.

**Fréchet Inception Distance** (FID, Heusel et al. 2017) measures the similarity between the output distribution of a generative image model and its training data. Rather than comparing the images directly, they are first encoded by an inception network. Rather than evaluating the output logits (see IS), the FID score is calculated as the squared Wasserstein distance between the image embeddings in the real and synthetic data. FID can be applied to individual frames in a video sequence to study the image quality of generative video models, but it fails to properly measure temporal coherence.

**Fréchet Video Distance** (FVD, Unterthiner et al. 2018) has been proposed as an extension of FID for the video domain. Its inception net is comprised of a 3D Convnet trained on action recognition tasks in Youtube videos. The authors demonstrate that the FVD measure is not only sensitive to spatial degradation (different kinds of noise), but also to temporal aberrations such as swapping of video frames. FVD is a commonly used metric for assessing the quality of unconditional or text-conditioned video generation.

**Kernel Video Distance (KVD)** Unterthiner et al. (2018) is an alternative to FVD. It is computed in an analogous manner, except that a polynomial kernel is applied to the features of the inception net. The authors found that FVD aligns better with human judgements than KVD. Nevertheless, both are commonly reported as benchmark metrics for unconditional video generation.

**Optical flow** describes the pixel-level displacement between neighboring video frames. It is estimated using neural networks, such as RAFT (Teed & Deng, 2020) or GMFlow (Xu et al., 2022). Optical flow estimates can be used to assess temporal consistency of an edited video sequence against a baseline. Commonly, the endpoint error (EPE) is used, which expresses the Euclidean distance between the estimated flow of the edited video and the original video.

### 5.4 Benchmarks

Commonly used evaluation datasets for video generation include UCF-101 (Soomro et al., 2012), MSR-VTT (Xu et al., 2016), Tai-Chi-HD (Siarohin et al., 2019), and Sky Time-Lapse (Radford et al., 2021). All four benchmarks can be calculated on samples that have been generated in an unconditional manner. For UCF-101, a second benchmark is sometimes reported on conditional generation where the 101 class labels are used for guiding the generative process. In this case, IS can be used as an evaluation metric. For MSR-VTT, conditional generation using the 200,000 human video annotations as text prompts can also be evaluated. Here, CLIP text-similarity is often reported as a measure of text-video alignment. Most often, the benchmarked models are either directly trained on the train split of the evaluation data set, or they are pre-trained on a separate large video data set (such as WebVid-10M or HD-Villa-100M) and later fine-tuned

Table 3: Overview of video diffusion models and their applications.

| Paper | Model | Application | Max. Resolution | Methodology | Shots |
|---|---|---|---|---|---|
| Ho et al. (2022c) | VDM | T V L | 128×128×64 | FA ↑S ↑T AR | Many |
| Singer et al. (2022) | Make-a-Video | T I V | 768×768×76 | FA ↑S ↑T | Many |
| Ho et al. (2022a) | ImagenVideo | T | 1280×768×128 | FA ↑S ↑T | Many |
| Zhou et al. (2022) | MagicVideo | T I V | 1024×1024×61 | P L FA ↑S ↑T | Many |
| Blattmann et al. (2023) | VideoLDM | T V | 2048×1280×90000 | P L FA ↑S ↑T AR | Many |
| Khachatryan et al. (2023) | Text2Video-Zero | T V | 512×512×8+ | P L | Many |
| Guo et al. (2023) | AnimateDiff | T | 512×512×16 | P L FA | Many |
| Chen et al. (2023) | MCDiff | I | 256×256×10 | L AR | Many |
| Yin et al. (2023) | Nuwa-XL | T L | NaN×NaN×1024 | P L FA ↑T | Many |
| He et al. (2022) | LVDM | T L | 256×256×1024 | P L FA ↑T AR | Many |
| Harvey et al. (2022) | FDM | V L | 128×128×15000 | P FA ↑T AR | Many |
| Lu et al. (2023) | VDT | I V | 256×256×30 | L FA ↑T | Many |
| Wang et al. (2023a) | Gen-L-Video | T V L | 512×512×hundreds | P L 3D | NaN |
| Zhu et al. (2023) | MovieFactory | T L | 3072×1280×NaN | P L FA ↑S | Many |
| Sun et al. (2023) | GLOBER | T L | 256×256×128 | P L 3D FA | Many |
| Luo et al. (2023) | VideoFusion | T L | 128×128×512 | P ↑S AR | Many |
| Hu et al. (2023) | GAIA-1 | T I L | 288×512×minutes | FA ↑T AR | Many |
| Lee et al. (2023a) | Soundini | A | 256×256×NaN | P | One |
| Lee et al. (2023b) | AADiff | I A | 512×512×150 | P L | Zero |
| Liu et al. (2023d) | Generative Disco | A | 512×512×NaN | P L | Zero |
| Tang et al. (2023) | Composable Diffusion | T I V A | 512×512×16 | P L FA | Many |
| Stypułkowski et al. (2023) | Diffused Heads | A | 128×128×8-9s | AR | Many |
| Zhua et al. (2023) | (Audio Heads) | A | 1024×1024×NaN | P ↑S AR | Many |
| Casademunt et al. (2023) | Laughing Matters | A | 128×128×50 | FA AR | Many |
| Molad et al. (2023) | Dreamix | I V | 1280×768×128 | P FA ↑S ↑T | One |
| Wu et al. (2022b) | Tune-A-Video | V | 512×512×100 | P L FA AR | One |
| Qi et al. (2023) | FateZero | V | 512×512×100 | P L FA AR | One |
| Liu et al. (2023b) | Video-P2P | V | 512×512×100 | P L FA AR | One |
| Ceylan et al. (2023) | Pix2Video | V | 512×512×NaN | P L FA AR | Zero |
| Esser et al. (2023) | Runway Gen-2 | T I V | 448×256×8 | P L FA | Many |
| Xing et al. (2023) | Make-Your-Video | V | 256×256×64 | P L FA AR | Many |
| Ma et al. (2023) | Follow Your Pose | V | 512×512×100 | P L FA AR | Many |
| Zhao et al. (2023) | Make-A-Protagonist | V | 768×768×8 | P L FA | One |
| Zhang et al. (2023) | ControlVideo | V | 512×512×100 | P L 3D ↑T | Zero |
| Wang et al. (2023b) | vid2vid-zero | V | 512×512×8 | P L AR | Zero |
| Huang et al. (2023) | Style-A-Video | V | 512×256×NaN | P L | Zero |
| Yang et al. (2023) | Rerender A Video | V | 512×512×NaN | P L ↑T AR | Zero |
| Liu et al. (2023a) | ColorDiffuser | V | 256×256×NaN | P L FA | Many |

T : txt2vid, I : img2vid, V : vid2vid, A : aud2vid, L : long vid

P : pre-trained model, L : latent space, 3D : full 3D attn./conv., FA : factorized attn./conv.,

↑S : spatial upsampling, ↑T : temporal upsampling, AR : auto-regressive

on the evaluation data set. However, some papers evaluate their model in a zero-shot setting, where the model has not been trained on the evaluation data set at all. These discrepancies between evaluation setups mean that a direct comparison of benchmark results across studies should be taken with a grain of salt.

The benchmark results for video generation are summarized in Table 4. Make-A-Video (Singer et al., 2022), one of the early diffusion-based video models, still holds state-of-the-art FVD and IS scores on the UCF-101 conditional generation benchmark. It not only outperforms all GAN and autogregressive models, but also the newer diffusion-based models. It is pre-trained on both the WebVid-10M and HD-Villa-100M data sets, which gives it an advantage in terms of the quantity of the training data over most other models. Make-A-Video also holds the best CLIP-similarity and FID scores in the zero-shot text-conditioned MSR-VTT benchmark. Make-A-Video is outperformed by Make-Your-Video (Xing et al., 2023) when it comes to zero-shot performance on UCF-101, although the latter uses depth maps as additional conditioning. Therefore, both models are not directly comparable. VideoFusion (Luo et al., 2023) has achieved the best FVD score on the unconditional Tai-Chi-HD and Sky Time-lapse benchmarks, as well as the best KVD score on Tai-Chi-HD. It is outperformed by LVDM (He et al., 2022) when it comes to KVD on the Sky Time-lapse benchmark. MagicVideo (Zhou et al., 2022) has the best FID score on UCF-101 and the best FVD score on MSR-VTT, although our comparison includes only few other datasets competing in those categories.

Table 4: Video generation benchmarks.

| Model | Resolution | Zero-Shot | Conditioning | Training | UCF-101 FID↓ | FVD↓ | IS↑ | MSR-VTT CLIP-Sim↑ | FID↓ | FVD↓ | Tai-Chi-HD FVD↓ | KVD↓ | Sky Time-lapse FVD↓ | KVD↓ |
|---|---|---|---|---|---|---|---|---|---|---|---|---|---|---|
| **GAN Models** | | | | | | | | | | | | | | |
| MoCoGAN (2018) | 64×64 | No | - | - | 26998 | - | 12.42 | - | - | - | - | - | - | - |
| TGAN-v2 (2020) | 64×64 | No | - | - | 3431 | - | 26.6 | - | - | - | - | - | - | - |
| TGAN-v2 (2020) | 128×128 | No | - | - | 3497 | - | 28.87 | - | - | - | - | - | - | - |
| TGAN-F (2020) | 64×64 | No | - | - | 8942 | - | 13.62 | - | - | - | - | - | - | - |
| TGAN-F (2020) | 128×128 | No | - | - | 7817 | - | 22.91 | - | - | - | - | - | - | - |
| DVD-GAN (2019) | 128×128 | No | Class | - | - | - | 32.97 | - | - | - | - | - | - | - |
| MoCoGAN-HD (2021) | 256×256 | No | - | UCF-101: test split incl. | - | 700 | 34 | - | - | - | 144.7 | 25.4 | 183.6 | 13.9 |
| DIGAN (2022) | 128×128 | No | - | - | - | 655 | 29.7 | - | - | - | 128.1 | 20.6 | 114.6 | 6.8 |
| DIGAN (2022) | 128×128 | No | - | UCF-101: test split incl. | - | 577 | 32.7 | - | - | - | 128.1 | 20.6 | 114.6 | 6.8 |
| **Transformer Models** | | | | | | | | | | | | | | |
| VideoGPT (2021) | 128×128 | No | - | - | - | - | 24.69 | - | - | - | - | - | - | - |
| NUWA (2022a) | 128×128 | No | - | VATEX | - | - | - | 0.24 | 47.7 | - | - | - | - | - |
| TATS-base (2022) | 128×128 | No | - | - | - | 420 | 57.6 | - | - | - | 94.6 | 8.8 | 132.6 | 5.7 |
| TATS-base (2022) | 128×128 | No | Class | - | - | 332 | 79.3 | - | - | - | - | - | - | - |
| CogVideo (Chinese) (2022) | 480×480 | Yes | Text | internal data | 185 | 751.3 | 23.6 | 0.26 | 24.8 | - | - | - | - | - |
| CogVideo (English) (2022) | 480×480 | Yes | Text | internal data | 179 | 701.6 | 25.3 | 0.26 | 23.6 | - | - | - | - | - |
| CogVideo (English) (2022) | 160×160 | Yes | Text | internal data | - | 626 | 50.5 | - | 49 | 1294 | - | - | - | - |
| **Diffusion Models** | | | | | | | | | | | | | | |
| VDM (2022c) | 64×64 | No | - | - | 295 | - | 57 | - | - | - | - | - | - | - |
| Make-a-Video (2022) | 256×256 | Yes | Text | WebVid10M, HD-VILA-10M | - | 367.2 | 33 | **0.3** | **13.2** | - | - | - | - | - |
| Make-a-Video (2022) | 256×256 | No | Text | WebVid10M, HD-VILA-10M | - | **81.3** | **82.6** | - | - | - | - | - | - | - |
| MagicVideo (2022) | 256×256 | Yes | Text | WebVid10M, HD-VILA-100M | **145** | 665 | - | - | 36.5 | **998** | - | - | - | - |
| LVDM (2022) | 256×256 | No | - | UCF-101: test split incl. | - | 372 | - | - | - | - | 99 | 15.3 | 95.2 | **3.9** |
| VideoLDM (SD 1.4) (2023) | 1280×2048 | Yes | Text | WebVid-10M | - | 656.5 | 29.5 | 0.29 | - | - | - | - | - | - |
| VideoLDM (SD 2.1) (2023) | 1280×2048 | Yes | Text | WebVid-10M | - | 550.6 | 33.5 | 0.29 | - | - | - | - | - | - |
| Make-Your-Video (2023) | 256×256 | Yes | Text + Depth | WebVid-10M | - | 330.5 | - | - | - | - | - | - | - | - |
| VDT (2023) | 64×64 | No | - | - | - | 225.7 | - | - | - | - | - | - | - | - |
| VideoFusion (2023) | 128×128 | No | - | - | - | 220 | 72.2 | - | - | - | **56.4** | **6.9** | 47 | 5.3 |
| VideoFusion (2023) | 128×128 | No | Text | - | - | 173 | 80 | - | - | - | - | - | - | - |
| GLOBER (2023) | 128×128 | No | - | - | - | 239.5 | - | - | - | - | 124.2 | - | - | - |
| GLOBER (2023) | 128×128 | No | Text | - | - | 151.5 | - | - | - | - | - | - | - | - |
| GLOBER (2023) | 256×256 | No | - | - | - | 252.7 | - | - | - | - | - | - | 78.1 | - |
| GLOBER (2023) | 256×256 | No | Text | - | - | 168.9 | - | - | - | - | - | - | - | - |

## 6 Video Generation

### 6.1 Unconditional Generation & Text-to-Video

Unconditional video generation and text-conditioned video generation are common benchmarks for generative video models. Prior to diffusion models, Generative Adversarial Networks (GANs, Goodfellow et al. 2014) and auto-regressive transformer models (Vaswani et al., 2017) have been popular choices for generative video tasks. In the following, we provide a short overview over a few representative GAN and auto-regressive transformer video models. We then introduce a selection of competing diffusion models starting in Sec. 6.1.3.

#### 6.1.1 GAN Video Models

**TGAN** (Saito et al., 2017) employs two generator networks: The temporal generator creates latent features that represent the motion trajectory of a video. This feature vector can be fed into an image generator

that creates a fixed number of video frames in pixel space. TGAN-v2 (Saito et al., 2020) uses a cascade of generator modules to create videos at various temporal resolutions, making the process more efficient. TGAN-F (Kahembwe & Ramamoorthy, 2020) is another improved version that relies on lower-dimensional kernels in the discriminator network.

**MoCoGAN** (Tulyakov et al., 2018) decomposes latent space into motion and content-specific parts by employing two separate discriminators for individual frames and video sequences. At inference time, the content vector is kept fixed while the next motion vector for each frame is predicted in an auto-regressive manner using a neural network. MoCoGAN was evaluated on unconditional video generation on the UCF-101 and Tai-Chi-HD datasets and achieved higher IS scores than the preceding TGAN and VGAN models.

**DVD-GAN** (Clark et al., 2019) uses a similar dual discriminator setup to MoCoGAN. The main difference is that DVD-GAN does not use auto-regressive prediction but instead generates all video frames in parallel. It outperformed previous methods such as TGAN-v2 and MoCoGAN on the UCF-101 dataset in terms of IS score, although DVD-GAN conditioned its generation on class labels, whereas the other approaches were unconditional.

**MoCoGAN-HD** (Tian et al., 2021) disentangles content and motion in a different way from the previous approaches. A motion generator is trained to predict a latent motion trajectory, which can then be passed as input to a fixed image generator. It outperformed previous approaches on unconditional generation in the UCF-101, Tai-Chi-HD, and Sky Time-lapse benchmarks.

**DIGAN** (Yu et al., 2022) introduces an implicit neural representation-based video GAN architecture that can efficiently represent long video sequences. It follows a similar content-motion split as discussed above. The motion discriminator judges temporal dynamics based on pairs of video frames rather than the whole sequence. These improvements enable the model to generate longer video sequences of 128 frames. DIGAN achieved state-of-the-art results on UCF-101 in terms of IS and FVD score, as well as on Sky Time-lapse and Tai-Chi-HD in terms of FVD and KVD scores.

### 6.1.2   Auto-Regressive Transformer Video Models

**VideoGPT** (Yan et al., 2021) uses a 3D VQ-VAE (Van Den Oord et al., 2017) to learn a compact video representation. An auto-regressive transformer model is then trained to predict the latent code of the next frame based on the preceding frames. While VideoGPT did not outperform the best GAN-based models at the time, namely TGAN-v2 and DVD-GAN, it achieved a respectable IS score on the UCF-101 benchmark considering its simple architecture.

**NÜWA** (Wu et al., 2022a) also uses a 3D VQ-VAE with an auto-regressive transformer generator. It is pre-trained on a variety of data sets that enable it to perform various generation and editing tasks in the video and image domains. Its text-conditioned video generation capability was evaluated on the MSR-VTT data set.

**TATS** (Ge et al., 2022) introduces several improvements that address the issue of quality degradation that auto-regressive transformer models face when generating long video sequences. It beat previous methods on almost all metrics for UFC-101 (unconditional and class-conditioned), Tai-Chi-HD, and Sky Time-lapse. Only DIGAN maintained a higher FVD score on the Sky Time-lapse benchmark.

**CogVideo** (Hong et al., 2022) is a text-conditioned transformer model. It is based on the pre-trained text-to-image model CogView2 (Ding et al., 2022), which is expanded with spatio-temporal attention layers. The GPT-like transformer generates key frames in a latent VQ-VAE space and a second upsampling model interpolates them to a higher framerate. The model was trained on an internal data set of 5.4 mio. annotated videos with a resolution of $160 \times 160$. It was then evaluated on the UCF-101 data set in a zero-shot setting by using the 101 class labels as text prompts. It beats most other models in terms of FVD and IS score except for TATS.

### 6.1.3 Diffusion Models

Producing realistic videos based on only a text prompt is one of the most challenging tasks for video diffusion models. A main problem lies in the relative lack of suitable training data. Publicly available video data sets are usually unlabeled, and human annotated labels may not even accurately describe the complex relationship between spatial and temporal information. Many authors therefore supplement training of their models with large data sets of labeled images or build on top of a pre-trained text-to-image model. The first video diffusion models (Ho et al., 2022c) had very high computational demands paired with relatively low visual fidelity. Both aspects have significantly been improved through architectural advancements, such as moving the denoising process to the latent space of a variational auto-encoder (He et al., 2022; Zhou et al., 2022) and using upsampling techniques such as CDMs (Ho et al. 2022a, see section 3.4).

Ho et al. (2022c) present an early diffusion-based video generation model called **VDM**. It builds on the 3D UNet architecture proposed by Çiçek et al. (2016), extending it by factorized spatio-temporal attention blocks. This produces videos that are 16 frames long and $64 \times 64$ pixels large. These low-resolution videos can then be extended to $128 \times 128$ pixels and 64 frames using a larger upsampling model. The models are trained on a relatively large data set of labeled videos as well as single frames from those videos, which enables text-guided video generation at time of inference. However, this poses a limitation of this approach since labeled video data is relatively difficult to come by.

Singer et al.'s (2022) **Make-a-Video** address this issue by combining supervised training of their model on labeled images with unsupervised training on unlabeled videos. This allows them to access a wider and more diverse pool of training data. They also split the convolution layers in their UNet model into 2D spatial convolutions and 1D temporal convolutions, thereby alleviating some of the computational burden associated with a full 3D Unet. Finally, they train a masked spatiotemporal decoder on temporal upsampling or video prediction tasks. This enables the generation of longer videos of up to 76 frames. Make-a-Video was evaluated on the UCF-101 and MSR-VTT benchmarks where it outperformed all previous GAN and autoregressive transformer models.

Ho et al. (2022a) use a cascaded diffusion process (Ho et al. 2022b, see Fig. 4) that can generate high resolution videos in their model called **ImagenVideo**. They start with a base model that synthesizes videos with $40 \times 24$ pixels and 16 frames, and upsample it over six additional diffusion models to a final resolution of $1280 \times 768$ pixels and 128 frames. The low-resolution base model uses factorized space-time convolutions and attention. To preserve computational resources, the upsampling models only rely on convolutions. ImagenVideo is trained on a large proprietary data set of labeled videos and images in parallel, enabling it to emulate a variety of visual styles. The model also demonstrates the ability to generate animations of text, which most other models struggle with.

Zhou et al.'s (2022) **MagicVideo** adapts the Latent Diffusion Models (Rombach et al. 2022, see Fig. 4) architecture for video generation tasks. In contrast to the previous models that operate in pixel space, their diffusion process takes place in a low-dimensional latent embedding space defined by a pre-trained variational auto-encoder (VAE). This significantly improves the efficiency of the video generation process. This VAE is trained on video data and can thereby reduce motion artefacts compared to VAEs used in text-to-image models. The authors use a pre-trained text-to-image model as the backbone of their video model with added causal attention blocks. The model is fine-tuned on data sets of labeled and unlabeled videos. It produces videos of $256 \times 256$ pixels and 16 frames that can be upsampled using separate spatial and temporal super resolution models to $1024 \times 1024$ pixels and 61 frames. In addition to text-to-video generation, the authors also demonstrate video editing and image animation capabilities of their model.

Blattmann et al. (2023) present another adaptation of the Latent Diffusion Models (Rombach et al., 2022) architecture to text-to-video generation tasks called **VideoLDM**. Similar to Zhou et al. (2022), they add temporal attention layers to a pre-trained text-to-image diffusion model and fine-tune them on labeled video data. They demonstrate that, in addition to text-to-video synthesis, their model is capable of generating long driving car video sequences in an auto-regressive manner, as well as of producing videos of personalized characters using Dreambooth (Ruiz et al., 2023).

## 6.2 Training-Free Models

Khachatryan et al.'s (2023) **Text2Video-Zero** completely eschews the need for video training data, instead relying only on a pre-trained text-to-image diffusion model that is augmented with cross-frame attention blocks. Motion is simulated by applying a warping function to latent frames, although it has to be mentioned that the resulting movement lacks realism compared to models trained on video data. Spatio-temporal consistency is improved by masking foreground objects with a trained object detector network and smoothing the background across frames. Similar to Zhou et al. (2022), the diffusion process takes place in latent space.

## 6.3 Personalized Generation

Personalized generation allows a user to adjust a pre-trained text-to-image model such that it learns concepts from a small set of personal images. Popular methods for this are model fine-tuning model (e.g. Dreambooth, Ruiz et al. 2023, LoRA, Hu et al. 2021), as well as textual inversion (Gal et al., 2022).

Guo et al. (2023) offer a text-to-video model developed with personalized image generation in mind. Their **AnimateDiff** extends a pre-trained Stable Diffusion model with a temporal adapter module merely containing self-attention blocks trained on video data. In this way, simple movement can be induced. The authors demonstrate that their approach is compatible with personalized image generation techniques such as Dreambooth (Ruiz et al., 2023) and LoRA (Hu et al., 2021).

## 6.4 Image-Conditioned Generation

There appear to be very few models that mainly focus on image animation, but several more generalized models offer this capability, such as Make-a-Video (Singer et al., 2022), MagicVideo (Zhou et al., 2022), Dreamix (Molad et al., 2023), Runway Gen-2 (Esser et al., 2023), AADiff (Lee et al., 2023b), and Composable Diffusion (Tang et al., 2023). Image animation is closely linked to video prediction in the sense that most of these models use the same masked prediction mechanism for both. Here, the input image is used as the first frame of a video sequence and generation of subsequent frames is conditioned on this information.

Chen et al. (2023) focus on the task of animating images in accordance with motion cues. Their Motion-Conditioned Diffusion Model (**MCDiff**) accepts an input image and lets the user indicate the desired motion by drawing strokes on top of it. The model then produces a short video sequence in which objects move in accordance with the motion cues. It can dissociate between foreground (e.g. actor movement) or background motion (i.e. camera movement), depending on the context. The authors use an auto-regressive approach to generate each video frame conditioned on the previous frame and predicted motion flow. For this, the input motion strokes are decomposed into smaller segments and passed to a UNet flow completion model to predict motion in the following frame. A denoising diffusion model receives this information and uses it to synthesize the next frame. The flow completion model and the denoising model are first trained separately but later fine-tuned jointly on unannotated videos.

# 7 Video Completion & Long Video Generation

Most video diffusion models can only generate a fixed number of video frames per sequence. In order to circumvent this limitation, auto-regressive extension and temporal upsampling methods have been proposed (see Section 4.2). Models adopting these methods often adjust and combine them in unique ways that benefit computational speed or consistency. A common problem of these approaches is that they tend to generate videos that suffer from repetitive content. Some models have therefore explored ways to generate videos with changing scenes by varying the text prompts over time.

## 7.1 Temporal Upsampling & Video Prediction

Yin et al.'s (2023) **NUWA-XL** model uses an iterative hierarchical approach to generate long video sequences of several minutes. It first generates evenly spaced key frames from separate text prompts that form a rough outline of the video. The frames in-between are then filled in with a local diffusion model conditioned on

two key frames. This process is applied iteratively to increase the temporal resolution with each pass. Since this can be parallelized, the model achieves much faster computation times than auto-regressive approaches for long video generation. The authors train the model on a new training data set consisting of annotated Flintstones cartoons. Simple temporal convolution and attention blocks are inserted into the pre-trained text-to-image model to learn temporal dynamics.

He et al. (2022) tackle the task of generating long videos with over 1,000 frames with their Long Video Diffusion Model (**LVDM**). It combines auto-regressive and hierarchical approaches for first generating long sequences of key frames and then filling in missing frames. In order to reduce quality degradation induced by auto-regressive sampling, the authors use classifier-free guidance and conditional latent perturbation which conditions the denoising process on noisy latents of reference frames. The model utilizes a dedicated video encoder and combines 2D spatial with 1D temporal self-attention. It can be used for unconditional video generation or text-to-video tasks.

Harvey et al. (2022) similarly explore methods for generating long video sequences with video models that have a fixed number of output frames. Their Flexible Diffusion Model (**FDM**) accepts an arbitrary number of conditioning frames to synthesize new frames, thereby allowing it to either extend the video in an auto-regressive manner or to use a hierarchical approach (similar to NUWA-XL, Yin et al. 2023). The authors explore variations of these sampling techniques and suggest an automated optimization routine that finds the best one for a given training data set.

Lu et al. (2023) propose Video Diffusion Transformer (**VDT**), a diffusion-based video model that uses a vision transformer architecture (Peebles & Xie, 2022). The reported advantages of this type of architecture over the commonly used UNet include the ability to capture long-range temporal dynamics, to accept conditioning inputs of varying lengths, and the scalability of the model. VDT was trained on more narrow data sets of unlabeled videos and accomplished tasks such as video prediction, temporal interpolation, and image animation in those restricted domains.

## 7.2 Alternative Approaches

Wang et al.'s (2023a) **Gen-L-Video** generates long video sequences by denoising overlapping shorter video segments in parallel. A video diffusion model predicts the denoised latent in each video segment individually. The noise prediction for a given frame is than aggregated through interpolation across all segments in which it appears. This leads to greater coherence across the long video sequence. The authors apply this new method to existing frameworks in the text-to-video (LVDM, He et al. 2022), tuning-free video-to-video (Pix2Video, Ceylan et al. 2023), and one-shot tuning video-to-video (Tune-A-Video, Wu et al. 2022b) domains.

Zhu et al. (2023) follow a unique approach for generating long video sequences in their **MovieFactory** model. Rather than extending a single video clip, they generate a movie-like sequence of separate related clips from a single text prompt. ChatGPT is used to turn the brief text prompt into ten detailed scene descriptions. Each scene description is then passed as a prompt to the video diffusion model to generate a segment of the video sequence. Finally, audio clips matching each video scene are retrieved from a sound data base. The pre-trained text-to-image model (Stable Diffusion 2.0) is first expanded by additional ResNet and attention blocks that are trained in order to produce wide-screen images. In a second training step, 1D temporal convolution and attention blocks are added to learn temporal dynamics.

Sun et al.'s (2023) **GLOBER** is a model for generating videos of arbitrary length that does not rely on auto-regressive or hierarchical approaches. Instead, it first uses a video KL-VAE auto-encoder to extract global 2D features from key frames. It then provides these global features along with arbitrary frame indices to a UNET diffusion model that can directly generate frames at those positions. To ensure the temporal coherence and realism of the generated frames, a novel adversarial loss is introduced. During training, an adversarial discriminator model receives pairs of video frames at random positions along with their indices and has to predict whether the frames both originated from the input video, or whether one or both were generated by the diffusion model. To enable inference, a generator model based on the Diffusion Transformer architecture (Peebles & Xie, 2022) is trained to produce global features that mimick those of the video encoder given text prompts. GLOBER surpasses several competing models in terms of FVD score, but it's main advantage is a much faster computation time compared to auto-regressive methods.

Luo et al. (2023) improve the temporal coherence in their **VideoFusion** model by decomposing the noise added during the forward diffusion process. A base noise component is shared across all frames and characterizes the content of the entire video. Meanwhile, a residual component is specific to each frame and is partially related to the motion of objects. This approach saves computational resources since a smaller residual generator denoising model can be used to estimate the residual noise for each frame, whereas the base noise has to be estimated only once for the entire video using a pre-trained image model. The pre-trained base generator is fine-tuned jointly with the residual generator.

Hu et al.'s (2023) **GAIA-1** is a hybrid transformer-diffusion model that can generate driving car videos conditioned on images, text, or action tokens that represent speed and movement trajectories. During training, it first uses a VQ-GAN to transform input video frames into discrete tokens. An auto-regressive transformer world model is used to predict the next token in the sequence based on all preceding tokens using causal masking. A diffusion-based video decoder then translates the tokens back to pixel space by denoising random noise patterns conditioned on the generated token sequence. The decoder is trained to enable flexible applications such as auto-regressive video generation and frame interpolation.

## 8 Audio-conditioned Synthesis

Multimodal synthesis might be the most challenging task for video diffusion models. A key problem lies in how associations between different modalities can be learned. Similar to how CLIP models (Radford et al., 2021) encode text and images in a shared embedding space, many models learn a shared semantic space for audio, text, and / or video through techniques such as contrastive learning (Chen et al., 2020).

### 8.1 Audio-conditioned Generation & Editing

Lee et al.'s (2023a) **Soundini** model enables local editing of scenic videos based on sound clips. A binary mask can be specified to indicate a video region that is intended to be made visually consistent with the auditory contents of the sound clip. To this end a sliding window selection of the sound clip's mel spectrogram is encoded into a shared audio-image semantic space. During training, two loss-functions are minimized to condition the denoising process on the embedded sound clips: The cosine similarity between the encoded audio clip and the image latent influences the generated video content, whereas the cosine similarity between the image and audio gradients is responsible for synchronizing the video with the audio signal. In contrast to other models, Soundini does not extend its denoising UNet to the video domain, only generating single frames in isolation. To improve temporal consistency, bidirectional optical flow guidance is used to warp neighboring frames towards each other.

Lee et al. (2023b) generate scenic videos from text prompts and audio clips with their Audio-Aligned Diffusion Framework (**AADiff**). An audio clip is used to identify a target token from provided text tokens, based on the highest similarity of the audio clip embedding with one of the text token embeddings. For instance, a crackling sound might select the word "burning". While generating video frames, the influence of the selected target token on the output frame is modulated through attention map control (similar to Prompt-to-Prompt, Hertz et al. 2022) in proportion to the sound magnitude. This leads to changes of relevant video elements that are synchronized with the sound clip. The authors also demonstrate that their model can be used to animate a single image and that several sound clips can be inserted in parallel. The model uses a pre-trained text-to-image model to generate each video frame without additional fine-tuning on videos or explicit modeling of temporal dynamics.

Liu et al.'s (2023d) **Generative Disco** provides an interactive interface to support creation of music visualizations. They are implemented as visual transitions between image pairs created with a diffusion model from user-specified text prompts. The interval in-between the two images is filled according to the beat of the music, using a form of interpolation that employs design patterns to cause shifts in color, subject or style, or set a transient video focus on subjects. A large language model can further assist the user with choosing suitable prompts. While the model is restricted to simple image transitions and is therefore not able to produce realistic movement, it highlights the creative potential of video diffusion models for music visualization.

Tang et al. (2023) present a model called **Composable Diffusion** that can generate any combination of output modalities based on any combination of input modalities. This includes text, images, videos, and sound. Encoders for the different modalities are aligned in a shared embedding space through contrastive learning. The diffusion process can then be flexibly conditioned on any combination of input modalities by linearly interpolating between their embeddings. A separate denoising diffusion model is trained for each of the output modalities and information between the modality-specific models is shared through cross-attention blocks. The video model uses simple temporal attention as well as the temporal shift method from An et al. (2023) to ensure consistency between frames.

### 8.2 Talking Head Generation

Stypułkowski et al. (2023) have developed the first diffusion model for generating videos of talking heads. Their model **Diffused Heads** takes a reference image of the intended speaker as well as a speech audio clip as input. The audio clip is divided into short chunks that are individually embedded through a pre-trained audio encoder. During inference, the reference image as well as the last two generated video frames are concatenated with the noisy version of the curent video frame and passed through a 2D UNet. Additionally, the denoising process is conditioned on a sliding window selection of the audio embeddings. The generated talking faces move their lips in sync with the audio and display realistic facial expressions.

Zhua et al. (2023) follow a similar approach, but instead of using a reference image, their model accepts a reference video that is transformed to align with the desired audio clip. Face landmarks are first extracted from the video, then encoded into eye blink embeddings and mouth movement embeddings. The mouth movements are aligned with the audio clip using contrastive learning. Head positions and eye blinks are encoded with a VAE, concatenated together with the synchronized mouth movement embeddings, and passed as conditioning information to the denoising UNet.

Casademunt et al. (2023) focus on the unique task of laughing head generation. Similar to Diffused Heads (Stypułkowski et al., 2023), the model takes a reference image and an audio clip of laughter to generate a matching video sequence. The model combines 2D spatial convolutions and attention blocks with 1D temporal convolutions and attention. This saves computational resources over a fully 3D architecture and allows it to process 16 video frames in parallel. Longer videos can be generated in an auto-regressive manner. The authors demonstrate the importance of using a specialized audio-encoder for embedding the laughter clips in order to generate realistic results.

## 9 Video Editing

Editing can mean a potentially wide range of operations such as adjusting the lighting, style, or background, changing, replacing, re-arranging, or removing objects or persons, modifying movements or entire actions, and more. To avoid having to make cumbersome specifications for possibly a large number of video frames, a convenient interface is required. To achieve this, most approaches rely on textual prompts that offer a flexible way to specify desired edit operations at a convenient level of abstraction and generality. However, completely unconstrained edit requests may be in conflict with desirable temporal properties of a video, leading to a major challenge of how to balance temporal consistency and editability (see Section 4.3). To this end, many authors have experimented with conditioning the denoising process based on preprocessed features of the input video. One-shot tuning methods first fine-tune their parameters on the ground truth video. This ensures that the video content and structure can be reconstructed with good quality. On the other hand, tuning-free methods are not fine-tuned on the ground truth video, which makes the editing computationally more efficient.

### 9.1 One-Shot Tuning Methods

Molad et al. (2023) present a diffusion video editing model called **Dreamix** based on the ImagenVideo (Ho et al., 2022a) architecture. It first downsamples an input video, adds Gaussian noise to the low resolution version, then applies a denoising process conditioned on a text prompt. The model is finetuned on each input video and follows the joint training objective of preserving the appearance of both the entire video and

individual frames. The authors demonstrate that the model can edit the appearance of objects as well as their actions. It is also able to take either a single input image or a collection of images depicting the same object and animate it. Like ImagenVideo (Ho et al., 2022a), Dreamix operates in pixel space rather than latent space. Together with the need to finetune the model on each video, this makes it computationally expensive.

Wu et al. (2022b) base their **Tune-A-Video** on a pre-trained text-to-image diffusion model. Rather than fine-tuning the entire model on video data, only the projection matrices in the attention layers are trained on a given input video. The spatial self-attention layer is replaced with a spatio-temporal layer attending to previous video frames, while a new 1D temporal attention layer is also added. The structure of the original frames is roughly preserved by using latents obtained with DDIM inversion as the input for the generation process. The advantages of this approach are that fine-tuning the model on individual videos is relatively quick and that extensions developed for text-to-image tasks such as ControlNet (Zhang & Agrawala, 2023) or Dreambooth (Ruiz et al., 2023) can be utilized. Several models have subsequently built upon the Tune-A-Video approach and improved it in different ways:

Qi et al. (2023) employ an attention blending method inspired by Prompt-to-Prompt (Hertz et al., 2022) in their **FateZero** model. They first obtain a synthetic text description of the middle frame from the original video through BLIP (Li et al., 2022) that can be edited by the user. While generating a new image from the latent obtained through DDIM inversion, they blend self- and cross-attention masks of unedited words with the original ones obtained during the inversion phase. In addition to this, they employ a masking operation that limits the edits to regions affected by the edited words in the prompt. This method improves the consistency of generated videos while allowing for greater editability compared to Tune-A-Video.

Liu et al. (2023b) also base their **Video-P2P** model on Tune-A-Video and similar to FateZero, they incorporate an attention tuning method inspired by Prompt-to-Prompt. Additionally, they augment the DDIM inversion of the original video by using Null-text inversion (Mokady et al., 2023), thereby improving its reconstruction ability.

## 9.2 Tuning-free Editing Methods

### 9.2.1 Depth-conditioned Editing

Ceylan et al.'s (2023) **Pix2Video** continues the trend of using a pre-trained text-to-image model as the backbone for video editing tasks. In contrast to the previous approaches, it however eliminates the need for fine-tuning the model on each individual video. In order to preserve the coarse spatial structure of the input, the authors use DDIM inversion and condition the denoising process on depth maps extracted from the original video. Temporal consistency is ensured by injecting latent features from previous frames into self-attention blocks in the decoder portion of the UNet. The projection matrices from the stock text-to-image model are not altered. Despite using a comparatively light-weight architecture, the authors demonstrate good editability and consistency in their results.

Esser et al.'s (2023) **Runway Gen-1** enables video style editing while preserving the content and structure of the original video. This is achieved on the one hand by conditioning the diffusion process on CLIP embeddings extracted from a reference video frame (in addition to the editing text prompt), and on the other hand by concatenating extracted depth estimates to the latent video input. The model uses 2D spatial and 1D temporal convolutions as well as 2D + 1D attention blocks. It is trained on video and image data in parallel. Predictions of both modes are combined in a way inspired by classifier-free guidance (Ho & Salimans, 2022), allowing for fine-grained control over the tradeoff between temporal consistency and editability. The successor model Runway Gen-2 (unpublished) also adds image-to-video and text-to-video capabilities.

Xing et al. (2023) extend a pre-trained text-to-image model conditioned on depth maps to video editing tasks in their **Make-Your-Video** model, similar to Pix2Video (Ceylan et al., 2023). They add 2D spatial convolution and 1D temporal convolution layers, as well as cross-frame attention layers to their UNet. A causal attention mask limits the number of reference frames to the four immediately preceding ones, as the authors note that this offers the best trade-off between image quality and coherence. The temporal modules are trained on a large unlabeled video data set (WebVid-10M, Bain et al. 2021).

### 9.2.2 Pose-conditioned Editing

Ma et al.'s (2023) **Follow Your Pose** conditions the denoising process in Tune-A-Video on pose features extracted from an input video. The pose features are encoded and downsampled using convolutional layers and passed to the denoising UNet through residual connections. The pose encoder is trained on image data, whereas the spatio-temporal attention layers (same as in Tune-A-Video) are trained on video data. The model generates output that is less bound by the source video while retaining relatively natural movement of subjects.

Zhao et al.'s (2023) **Make-A-Protagonist** combines several expert models to perform subject replacement and style editing tasks. Their pipeline is able to detect and isolate the main subject (i.e. the "protagonist") of a video through a combination of Blip-2 (Li et al., 2023) interrogation, Grounding DINO (Liu et al., 2023c) object detection, Segment Anything (Kirillov et al., 2023) object segmentation, and XMem (Cheng & Schwing, 2022) mask tracking across the video. The subject can then be replaced with that from a reference image through Stable Diffusion inpainting with ControlNet depth map guidance. Additionally, the background can be changed based on a text prompt. The pre-trained Stable Diffusion UNet model is extended by cross-frame attention and fine-tuned on frames from the input video.

### 9.2.3 Multi-conditional Editing

Zhang et al.'s (2023) **ControlVideo** model extends ControlNet (Zhang & Agrawala, 2023) to video generation tasks. ControlNet encodes preprocessed image features using an auto-encoder and passes them through a fine-tuned copy of the first half of the Stable Diffusion UNet. The resulting latents at each layer are then concatenated with the corresponding latents from the original Stable Diffusion model during the decoder portion of the UNet to control the structure of the generated images. In order to improve the spatio-temporal coherence between video frames, ControlVideo adds full cross-frame attention to the self-attention blocks of the denoising UNet. Furthermore, it mitigates flickering of small details by interpolating between alternating frames. Longer videos can be synthesized by first generating a sequence of key frames and then generating the missing frames in several batches conditioned on two key frames each. In contrast to other video-to-video models that rely on a specific kind of preprocessed feature, ControlVideo is compatible with all ControlNet models, such as Canny or OpenPose. The pre-trained Stable Diffusion and ControlNet models also do not require any fine-tuning.

### 9.2.4 Other Approaches

Wang et al. (2023b) also adapt a pre-trained text-to-image model to video editing tasks without fine-tuning. Similar to Tune-A-Video and Pix2Video, their **vid2vid-zero** model replaces self-attention blocks with cross-frame attention without changing the transformation matrices. While the cross-frame attention in those previous models is limited to the first and immediately preceding frame, Wang et al. extend attention to the entire video sequence. Vid2vid-zero is not conditioned on structural depth maps, instead using a traditional DDIM inversion approach. To achieve better alignment between the input video and user-provided prompt, it optimizes the null-text embedding used for classifier-free guidance.

Huang et al. (2023) present **Style-A-Video**, a model aimed at editing the style of a video based on a text prompt while preserving its content. It utilizes a form of classifier-free guidance that balances three separate guidance conditions: CLIP embeddings of the original frame preserve semantic information, CLIP embeddings of the text prompt introduce stylistic changes, while CLIP embeddings of thresholded affinity matrices from self-attention layers in the denoising UNet encode the spatial structure of the image. Flickering is reduced through a flow-based regularization network. The model operates on each individual frame without any form of cross-frame attention or fine-tuning of the text-to-image backbone. This makes it one of the lightest models in this comparison.

Yang et al. (2023) also use ControlNet for spatial guidance in their **Rerender A Video** model. Similar to previous models, sparse causal cross-frame attention blocks are used to attend to an anchor frame and the immediately preceding frame during each denoising step. During early denoising steps, frame latents are additionally interpolated with those from the the anchor frame for rough shape guidance. Furthermore,

the anchor frame and previous frame are warped in pixel space to align with the current frame, encoded, and then interpolated in latent space. To reduce artefacts associated with repeated encoding, the authors estimate the encoding loss and shift the encoded latent along the negative gradient of the loss function to counteract the degradation. A form of color correction is finally applied to ensure color coherence across frames. This pipeline is used to generate key frames that are then filled in using patch-based propagation. The model produces videos that look fairly consistent when showing slow moving scenes but struggles with faster movements due to the various interpolation methods used.

### 9.3 Video Restoration

Liu et al. (2023a) present **ColorDiffuser**, a model specialized on colorization of grayscale video footage. It utilizes a pre-trained text-to-image model and specifically trained adapter modules to colorize short video sequences in accordance with a text prompt. Color Propagation Attention computes affinities between the current grayscale frame as Query, the reference grayscale frame as Key, and the (noisy) colorized reference frame latent as Value. The resulting frame is concatenated with the current grayscale frame and fed into a Coordinator Module that follows the same architecture as the Stable Diffusion UNet. Feature maps from the Coordinator module are then injected into the corresponding layers of the denoising UNet to guide the diffusion process (similar to ControlNet). During inference, an alternating sampling strategy is employed, whereby the previous and following frame are in turn used as reference. In this way, color information can propagate through the video in both temporal directions. Temporal consistency and color accuracy is further improved by using a specifically trained vector-quantized variational auto-encoder (VQVAE) that decodes the entire denoised latent video sequence.

### 9.4 Benchmarks

## 10 Outlook and Challenges

Video diffusion models have already demonstrated impressive results in a variety of use cases. However, there are still several challenges that need to be overcome before we arrive at models capable of producing longer video sequences with good temporal consistency.

One issue is the relative lack of suitable training data. While there are large data sets of labeled images that have been scraped rom the internet (Sec. 5.2), the available labeled video data are much smaller in size (Sec. 5.1). Many authors have therefore reverted to training their models jointly on labeled images and unlabeled videos or fine-tuning a pre-trained text-to-image model on unlabeled video data. While this compromise allows for learning of diverse visual concepts, it may not be ideal for capturing object-specific motion. One possible solution is to manually annotate video sequences (Yin et al., 2023), although it seems unlikely that this can be done on the scale required for training generalized video models. It is to be hoped that in the future automated annotation methods will develop that allow for generation of accurate video descriptions (Zare & Yazdi, 2022).

An even more fundamental problem is that simple text labels are often inadequate for describing the temporally evolving content of videos. This hampers the ability of current video models to generate more complex sequences of events. For this reason, it might be beneficial to examine alternative ways to describe video contents that represent different aspects more explicitly, such as the actors, their actions, the setting, camera angle, lighting, scene transitions, and so on.

A different challenge lies in the modeling of (long-term) temporal dependencies. Due to the memory limitations of current graphics cards, video models can typically only process a fixed number of video frames at a time. To generate longer video sequences, the model is extended either in an auto-regressive or hierarchical fashion, but this usually introduces artefacts or leads to degraded image quality over time. Possible improvements could be made on an architectural level. Most video diffusion models build on the standard UNet architecture of text-to-image models. To capture temporal dynamics, the model is extended by introducing cross-frame convolutions and / or attention. Using full 3D spatio-temporal convolutions and attention blocks are however prohibitively expensive. Many models therefore have adopted a factorized pseudo-3D architecture, whereby a 2D spatial block is followed by a 1D temporal block. While this compromise seems

necessary in the face of current hardware limitations, it stands to reason that full 3D architectures might be better able to capture complex spatio-temporal dynamics once the hardware allows it. In the meantime, other methods for reducing the computational burden of video generation will hopefully be explored. This could also enable new applications of video diffusion, such as real-time video-to-video translation.

## 11 Conclusion

In this survey, we have explored the current literature on video diffusion models. We have first categorized possible applications based on input modalities. Next, we have discussed technical aspects regarding the choice of architecture, modeling of temporal dynamics, and model training. Developments in the field have been outlined through paper summaries. We have concluded with remaining issues and potential for future improvements.

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
