# OpenReview forum: "Video Diffusion Models - A Survey"
_TMLR — Rejected by TMLR_

### Review · Reviewer_B2ge · 2023-11-19

**Summary Of Contributions:**

This survey provides a broad overview of recent advances in video diffusion models, organizing related works into relevant subtask categories. The paper covers several important aspects including applications, architectures, temporal dynamics, and evaluation metrics.

**Audience:**

No

**Claims And Evidence:**

Yes

**Requested Changes:**

- Involve discussions on datasets and performance comparisons.
- Provide illustrations to make the understanding easier.
- More detailed analysis, conclusion and insights should be included.

**Strengths And Weaknesses:**

Pros:
- The paper covers application taxonomy, model architecture, temporal dynamics, Evaluation Metrics, and provides an introduction for readers who want to know the overview of the video diffusion field.
- Table1 provides a systematic overview of prior works and figure 2 provides a good taxonomy of video diffusion applications.

Cons:
- While this survey provides a broad overview of video generation models, community could benefit much more from this paper when additional detailed conclusions and insights are provided.
- The datasets and performance comparisons are not discussed in this survey.
- In the Temporal Dynamics section (Sec 4), explicitly referencing key papers when describing different techniques would better highlight the most impactful contributions in this field. Additionally, including visual examples of model failures would aid reader comprehension. For example,  providing some unwanted object warping illustration for the description “In practice, this method in itself is not sufficient , … This can lead to unwanted object warping across”.
- The disadvantages behind training from scratch on video data (Sec 4.4) requires further elaboration to fully justify the training paradigm choice. Specifically, the inconsistent alignments between text and video compared to text and images needs to be explored.
- I agree that human evaluations are critical for assessing video generation quality. To emphasize their importance, the paper could compare and contrast human judgments with automated metrics, discussing the advantages and limitations of each. Standard best practices for human studies like detailed experimental design, principles for metric selection, and statistical significance testing should also be covered.
- While listing common quantitative metrics is helpful, their discussions lack depth and clear conclusions. For example, the statements on FVD's suitability for different tasks require supporting evidence and analysis. And the overview of optical flow EPE does not provide any meaningful takeaways for the reader. Expanding on these metrics with rigorous comparisons on relevant datasets would strengthen these sections.

---

> ### Author Response · Authors · 2024-01-02
> **Revision**
>
> Dear reviewer,
>
> thank you very much for your time and thoughtful feedback.
>
>
> A version of the updated paper where the main revisions have been marked in blue can be found in the supplementary material. We added tables 1, 2, 4.
>
> >While this survey provides a broad overview of video generation models, community could benefit much more from this paper when additional detailed conclusions and insights are provided.
>
> >More detailed analysis, conclusion and insights should be included.
>
> We have added an overview of benchmarks (Section 5.4, Table 4) on (un)conditional video generation on commonly used data sets. This comparison also includes non-diffusion methods that are discussed in section 6.1.1 and 6.1.2.
>
> >The datasets and performance comparisons are not discussed in this survey.
>
> >Involve discussions on datasets and performance comparisons.
>
> In addition to the benchmarks, we have added a sections on commonly used video (5.1) and image (5.2) training data sets, including tables (1, 2) that summarize their most salient aspects
>
> > In the Temporal Dynamics section (Sec 4), explicitly referencing key papers when describing different techniques would better highlight the most impactful contributions in this field. Additionally, including visual examples of model failures would aid reader comprehension. For example, providing some unwanted object warping illustration for the description “In practice, this method in itself is not sufficient , … This can lead to unwanted object warping across”.
>
> >Provide illustrations to make the understanding easier.
>
> We have added more references to key papers in the temporal dynamics section (4) as the methods part in general.
>
> >The disadvantages behind training from scratch on video data (Sec 4.4) requires further elaboration to fully justify the training paradigm choice. Specifically, the inconsistent alignments between text and video compared to text and images needs to be explored.
>
> We have expanded the discussion on why training on videos is often augmented with image data sets (Section 5), in addition to new sections that discuss representative video (5.1) and image (5.2) data sets.
>
> >I agree that human evaluations are critical for assessing video generation quality. To emphasize their importance, the paper could compare and contrast human judgments with automated metrics, discussing the advantages and limitations of each. Standard best practices for human studies like detailed experimental design, principles for metric selection, and statistical significance testing should also be covered.
>
> We agree that discussing methodological aspects of human studies would provide value for prospective readers. Unfortunately, we believe that we lack the necessary expertise in this domain to provide an insightful discussion.
>
> >While listing common quantitative metrics is helpful, their discussions lack depth and clear conclusions. For example, the statements on FVD's suitability for different tasks require supporting evidence and analysis. And the overview of optical flow EPE does not provide any meaningful takeaways for the reader. Expanding on these metrics with rigorous comparisons on relevant datasets would strengthen these sections.
>
> The sections on (evaluation) data sets and benchmarks hopefully address part of you criticism. We can not provide a deeper discussion of the pros and cons of different evaluation metrics at this time.

---

### Review · Reviewer_Pz4A · 2023-11-24

**Summary Of Contributions:**

The paper presents a survey on video diffusion methods for video generation and editing that have been developed in the last two years. It introduces a taxonomy of applications (Section 2), discusses architectural variants of image diffusion models (Section 3), outlines how those can be extended to video diffusion (Section 4), and discusses evaluation metrics (Section 5). The second part of the survey (Section 6) focuses on a literature review on text-to-video, image-to-video, and video-video, as well as multimodal synthesis and long video generation.

**Audience:**

Yes

**Broader Impact Concerns:**

no concerns

**Claims And Evidence:**

No

**Requested Changes:**

The paper requires a major revision in which the literature review is better integrated into the first paper (see above).
Besides that, addressing the points  list above under weaknesses would be needed.

**Strengths And Weaknesses:**

Strengths:
 + Video synthesis is a topic of increasing popularity, lacking a solid survey on recent trends with diffusion models so far (there is a concurrent work from Xing et al., https://arxiv.org/abs/2310.10647). This survey should thus be of significant interest to the TMLR audience.
 + The survey covers a large number of relevant works and summarizes them in Section 6. This will be helpful for practitioners and researchers for getting an overview of the current state of the field. Specifically, Table 1 is a valuable resource (however, it should be better discussed in the text, for instance many/one/zero-shot is not sufficiently detailed)
 +  The authors provide helpful illustrations in the figures that support understanding the concepts discussed in the survey.

Weaknesses:
 -  Organization of the paper: the first half (page 1-10) contains very few references to relevant work. Many statements are not supported by appropriate citations. The relevant works are then discussed in the second half (page 11-17), but this disconnect is suboptimal for a survey. Ideally, the first and second half should be merged in a major revision of the paper such that the literature overview is naturally blended with the discussions of taxonomy, architectures, datasets, and tasks.
 - Scope: The paper should also discuss works from "unconditional video synthesis" and  "video completion". Moreover, it should relate diffusion-based approaches to prior, non-diffusion based approaches such as GANs (that "they have quickly overtaken GAN-based approaches in popularity" is not a sufficient discussion) .  Also the related field of "video understanding with diffusion models" could be mentioned.
 - Phrasing: the authors repeatedly use phrases like "it seems to be the case" or "it might be". A survey should not reflect subjective impressions or speculations (as indicated by these phrases) but well supported observations about a field. The corresponding statements should be rephrased (if they are well supported observations) or be removed (if too subjective). Moreover, the introduction lacks a critical distance to the topic but "buys into" the hype with phrases like "tremendous ability" or "progress is being made on a daily basis".
 - Since arguably the most important ingredient for a video diffusion model is its training data, a survey should discuss existing datasets and their properties in detail. This is missing at the moment.
 - While difficult, it would still be desirable to compare different approaches quantitatively on a benchmark, in terms of metrics such as FVD.
 - Figure 1: the "architecture" dimension conflates the denoising architecture (U-Net vs. ViT) with the diffusion approach (latent vs. cascaded)
 - a general introduction of diffusion models including some formulas would be helpful for readers that do not yet have a background in the field.  Section 3.1 is too high-level for this in its current form.
 - Section 3.3: The linear output projection is missing in the self-attention. Moreover, "Even though far less commonly used, this approach might have distinct advantages, such as lower computational costs and, in the case of generative video models, more flexibility in regard to the length of the generated video sequence." Could the authors explain why this might be the case?
 - Section 3.5: citing only a single work (Rombach et al., 2022) for LDMs and no follow-up work is too shallow.
 - Figure 8 lacks a clear message. It shows different settings that differ in various ways; why are those grouped into a single figure?
 - "The task of editing an existing video can be viewed as less demanding than text-to-video generation." An explanation why it can be viewed that way would be desirable.
 - The section on "Long Video Generation" might also include hybrid auto-regressive/diffusion approaches such as GAIA-1 (https://arxiv.org/abs/2309.17080)
 - "While labeled images can be be scraped in large quantity from the internet, there is no comparable source of labeled video data." Is this statement true? There a millions of videos on the internet on platforms like YouTube that come with a summary text. They might be problematic in terms of usage license, but suchchallenges should be discussed in more detail.

---

> ### Author Response · Authors · 2024-01-02
> **Revisions (part 1)**
>
> Dear reviewer,
>
> thank you very much for your time and thoughtful feedback.
>
> A version of the updated paper where the main revisions have been marked in blue can be found in the supplementary material. Unfortunately, we were not able to mark the new tables.
>
> >Organization of the paper: the first half (page 1-10) contains very few references to relevant work. Many statements are not supported by appropriate citations. The relevant works are then discussed in the second half (page 11-17), but this disconnect is suboptimal for a survey. Ideally, the first and second half should be merged in a major revision of the paper such that the literature overview is naturally blended with the discussions of taxonomy, architectures, datasets, and tasks.
>
> >Phrasing: the authors repeatedly use phrases like "it seems to be the case" or "it might be". A survey should not reflect subjective impressions or speculations (as indicated by these phrases) but well supported observations about a field. The corresponding statements should be rephrased (if they are well supported observations) or be removed (if too subjective). Moreover, the introduction lacks a critical distance to the topic but "buys into" the hype with phrases like "tremendous ability" or "progress is being made on a daily basis".
>
> We substantiated our claims in the first part of the survey with appropriate citations. We have added cross-references to make comprehensive understanding easier.
>
>
> >Scope: The paper should also discuss works from "unconditional video synthesis" and "video completion". Moreover, it should relate diffusion-based approaches to prior, non-diffusion based approaches such as GANs (that "they have quickly overtaken GAN-based approaches in popularity" is not a sufficient discussion) . Also the related field of "video understanding with diffusion models" could be mentioned.
>
> We added more information towards unconditional video synthesis and provided some relevant benchmarks (see Fig. 4, Section 5.4). We find it difficult to completely separate it from text-conditioned synthesis, since most of the relevant papers deal with both.
>
>
> >Since arguably the most important ingredient for a video diffusion model is its training data, a survey should discuss existing datasets and their properties in detail. This is missing at the moment.
>
> We have added a new section on representative training data sets (Sections 5.1, 5.2).
>
>
> >While difficult, it would still be desirable to compare different approaches quantitatively on a benchmark, in terms of metrics such as FVD.
>
> We have added a section with benchmarks on unconditional and text-conditioned video generation (see Fig. 4, Section 5.4).
>
>
> >Figure 1: the "architecture" dimension conflates the denoising architecture (U-Net vs. ViT) with the diffusion approach (latent vs. cascaded)
>
> We have made a clearer separation between the denoising architecture and method for increasing resolution (CDM, LDM) in Fig. 1 and other parts of the review.
>
>
> >a general introduction of diffusion models including some formulas would be helpful for readers that do not yet have a background in the field. Section 3.1 is too high-level for this in its current form.
>
> We have expanded the methods section on DDPMs (3.1) to include the formal foundations (mainly adapted from Ho et al, 2020).
>
>
> >Section 3.3: The linear output projection is missing in the self-attention. Moreover, "Even though far less commonly used, this approach might have distinct advantages, such as lower computational costs and, in the case of generative video models, more flexibility in regard to the length of the generated video sequence." Could the authors explain why this might be the case?
>
> The ViT section (3.3) now includes a mention of the linear output projection for multi-head attention. We have removed the mention of lower computational cost as it lacks suitable evidence. We have added an explanation that the flexible length of generated video sequences is due to the auto-regressive nature of token predictions in transformer models at time of inference.
>
>
> >Section 3.5: citing only a single work (Rombach et al., 2022) for LDMs and no follow-up work is too shallow.
>
> We have added a brief mention of follow-up work on LDM models to section 3.5.
>
>
> >Figure 8 lacks a clear message. It shows different settings that differ in various ways; why are those grouped into a single figure?
>
> We have simplified figure 8 to focus on the main aspect we wanted to convey.

---

> ### Author Response · Authors · 2024-01-02
> **Revision (part 2)**
>
> >"The task of editing an existing video can be viewed as less demanding than text-to-video generation." An explanation why it can be viewed that way would be desirable.
>
> We have removed this line. Upon reflection, it is not clear to us whether video editing is indeed less demanding than text-conditioned generation, especially in cases that go beyond simple style editing.
>
>
> >The section on "Long Video Generation" might also include hybrid auto-regressive/diffusion approaches such as GAIA-1 (https://arxiv.org/abs/2309.17080)
>
> We have added it to our review.
>
> > "While labeled images can be be scraped in large quantity from the internet, there is no comparable source of labeled video data." Is this statement true? There a millions of videos on the internet on platforms like YouTube that come with a summary text. They might be problematic in terms of usage license, but suchchallenges should be discussed in more detail.
>
> We have adjusted the wording to reflect that the main problem lies in the size of the publicly available video data sets when compared to image data.

---

### Review · Reviewer_KyiL · 2023-12-19

**Summary Of Contributions:**

This paper provides a survey for video diffusion models. Authors focus on the applications, architecture, and temporal dynamics, followed by brief summaries of relevant papers.

**Audience:**

Yes

**Broader Impact Concerns:**

The reviewer did not see any major ethical concerns in this paper.

**Claims And Evidence:**

Yes

**Requested Changes:**

See the weakness above.

1. Provide the quantitive analysis of different models which can be categorized by different model architectures.

2. Add the video editing and understanding tasks to the evaluation section.

3. Provide more comparisons or figures in Section 6 instead of plain text for the literature review.

4. Provide the statistics of current pretraining datasets.

**Strengths And Weaknesses:**

Strengths:

1. The paper is well written and easy to follow.
2. The figures, such as Figure 4, 6 and 7, provide an extensive analysis of current model architectures.

Weaknesses:

1. The main concern is that there is no quantitive analysis about different models, such as the comparisons of FVD on different benchmarks.

2. In Section 6, there is a literature overview of different methods. Those methods are not well categorized based on different aspects. It is very difficult to understand the differences from the text.

3. For the evaluation section, there are a lot of missing details. Such as Inception score is often applied in evaluation on UCF-101. Besides to  video generation, video editing and video understanding tasks are also used to evaluate the quality of diffusion models.

4. It will be helpful if there is an analysis of the pretraining datasets.

---

> ### Author Response · Authors · 2024-01-02
> **Added Benchmarks and re-organized paper summaries**
>
> Dear reviewer,
>
> thank you very much for your time and thoughtful feedback.
>
> > The main concern is that there is no quantitive analysis about different models, such as the comparisons of FVD on different benchmarks.
> > Provide the quantitive analysis of different models which can be categorized by different model architectures.
>
> We have added an overview of quantitative benchmarks on the UCF101, MSR-VTT, Tai-Chi-HD, and Sky Time-lapse data sets for different models. See Table 4 and Chapter 5.4.
>
>
> > In Section 6, there is a literature overview of different methods. Those methods are not well categorized based on different aspects. It is very difficult to understand the differences from the text.
> > Provide more comparisons or figures in Section 6 instead of plain text for the literature review.
>
> We have re-organized the literature summaries according to more fine-grained criteria in order to ease understanding. The improved taxonomy is shown in Figure 1 and Figure 2, which contain links to the relevant chapters.
>
>
> > For the evaluation section, there are a lot of missing details. Such as Inception score is often applied in evaluation on UCF-101. Besides to video generation, video editing and video understanding tasks are also used to evaluate the quality of diffusion models.
>
> We have added descriptions of inception score and KVD to the evaluation section.
>
>
> > It will be helpful if there is an analysis of the pretraining datasets.
> > Provide the statistics of current pretraining datasets.
>
> We have added a section on commonly used training data sets, including a table that summarizes their most salient aspects. See Chapter 5.1 and 5.2, as well as Table 1 and 2.
>
>
> > Add the video editing and understanding tasks to the evaluation section.
>
> We have decided against including a section on video understanding, as it is beyond the focus of our attention. Video editing is briefly mentioned in section 5.3.

---

### Decision · Action_Editor_cwjM · 2024-02-03

**Recommendation:** Reject

**Comment:**

Ultimately, after the author responses, two of the reviewers lean towards rejection and one towards acceptance.

Reviewer Pz4A (leaning towards rejection) considers the paper improved but still unfinished with e.g. an empty section; the reviewer feels that due to the substantial revision a resubmission and independent review would be needed.
Similarly, reviewer KyiL (learning towards reject) feels that many details of data and analysis are missing and a major revision would be needed. Reviewer B2ge (learning towards acceptance) considers the survey beneficial to the community but suggests including recent works like VideoPoet.

In more detail, the reviewers saw several positive aspects:
+ The video synthesis topic was consided popular and lacking a survey, thus having interest to the TMLR audience [Pz4A]
+ Coverage of taxonomy, model architecture, temporal dynamics, and metrics was appreciated [B2ge], and the systematic overview and taxonomy were appreciated [B2ge].
+ The survey was considered to cover a large number of relevant works [Pz4A]
+ The paper was considered well written [KyiL]
+ The figures were considered helpful [KyiL,Pz4A]

However, many weaknesses were identified:
- Organization of the paper was criticized, with lack of references in the first half [Pz4A]. Authors added citations.
- Subjective writing was criticized [Pz4A]; authors added citations.
- An introduction to diffusion models was desired [Pz4A]; authors added material to one section.
- Poor categorization of methods in the literature review was criticized [KyiL]. Authors reorganized the section and provided an improved taxonomy.
- Scope of the paper was criticized. Discussion of video completion and unconditional video synthesis was desired, and relation of diffusion and non-diffusion based approaches [Pz4A]. Similarly, video editing [KyiL] and understanding tasks [KyiL,Pz4A] were also desired. Authors added some information on video synthesis, but chose not to add video understanding.
- Discussion of hybrid auto-regressive/diffusion approaches was desired [Pz4A]; authors added some details.
- Discussion of linear output projection in self-attention was desired [Pz4A]; authors added discussion of it and clarified some related statements.
- Additional detail on issues with training from scratch and inconsistent alignments was desired [B2ge]; authors added some discussion of augmenting with image data sets, it is unclear if the inconsistent alignment part was addressed.
- Additional references on temporal dynamics and visual examples of model failures were desired [B2ge]; authors added references, but no mention of model failure related improvements was stated.
- Discussion of training data was desired [Pz4A,B2ge], and a claim about lack of availability of labeled video data was criticized [Pz4A]; authors added a section on data and modified the statement.
- Lack of quantitative analysis and comparisons of different models on benchmarks was criticized [KyiL,Pz4A,B2ge]. Authors added an overview of FVD benchmark comparisons.
- Additional detailed conclusions and insights were desired [B2ge]; authors mentioned their added material on data although it is not clear if this answered the reviewer's concern.
- In-depth conclusions about quantitative metrics and comparing them on relevant data sets was desired [B2ge]; authors referred to their added material on benchmarks but declined to give deeper discussion of pros and cons of the evaluation metrics.
- Comparison of automated metrics with human judgments and best practices for human studies was desired [B2ge]; authors declined to address this citing lack of expertise.
- Additional details for the evaluation section were desired [KyiL]. Authors added some details.

Overall, the reviews seem to clearly indicate that part of the TMLR audience would be interested in the content of the paper. On the other hand, the reviews indicate the original version of the paper did not provide sufficiently convincing, comprehensive and clear information for a survey of this kind. Authors have made a large number of modifications in response to reviewer concerns, but declined to address some of them, and the quality of the improvement may need a full new review. Thus, at present I am recommending a rejection of this version but am encouraging authors to resubmit a major revision.

**Audience:**

Overall, the reviews seem to clearly indicate that part of the TMLR audience would be interested in the content of the paper.

**Claims And Evidence:**

The reviews indicate the original version of the paper did not provide sufficiently convincing, comprehensive and clear information for a survey of this kind. Authors have made a large number of modifications in response to reviewer concerns, but declined to address some of them, and the quality of the improvement may need a full new review.

The majority of reviewers feel similarly: Reviewer Pz4A (leaning towards rejection) considers the paper improved but still unfinished with e.g. an empty section; the reviewer feels that due to the substantial revision a resubmission and independent review would be needed. Similarly, reviewer KyiL (learning towards reject) feels that many details of data and analysis are missing and a major revision would be needed. Reviewer B2ge (learning towards acceptance) considers the survey beneficial to the community but suggests including recent works like VideoPoet.

**Resubmission Of Major Revision:**

The authors may consider submitting a major revision at a later time.